# Dermal appendage-dependent patterning of zebrafish *atoh1a+* Merkel cells

**Tanya L Brown**[1†]**, Emma C Horton**[1†]**, Evan W Craig**[1]**, Camille EA Goo**[1]**, Erik C Black**[1,2]**, Madeleine N Hewitt**[2,3]**, Nathaniel G Yee**[1]**, Everett T Fan**[1]**, David W Raible**[3,4,5]**, Jeffrey P Rasmussen**[1,5]*****

[1]Department of Biology, University of Washington, Seattle, United States; [2]Molecular and Cellular Biology Program, University of Washington, Seattle, United States; [3]Department of Biological Structure, University of Washington, Seattle, United States; [4]Department of Otolaryngology - Head and Neck Surgery, University of Washington, Seattle, United States; [5]Institute for Stem Cell and Regenerative Medicine, University of Washington, Seattle, United States

**Abstract** Touch system function requires precise interactions between specialized skin cells and somatosensory axons, as exemplified by the vertebrate mechanosensory Merkel cell-neurite complex. Development and patterning of Merkel cells and associated neurites during skin organogenesis remain poorly understood, partly due to the in utero development of mammalian embryos. Here, we discover Merkel cells in the zebrafish epidermis and identify Atonal homolog 1a (Atoh1a) as a marker of zebrafish Merkel cells. We show that zebrafish Merkel cells derive from basal keratinocytes, express neurosecretory and mechanosensory machinery, extend actin-rich microvilli, and complex with somatosensory axons, all hallmarks of mammalian Merkel cells. Merkel cells populate all major adult skin compartments, with region-specific densities and distribution patterns. In vivo photoconversion reveals that Merkel cells undergo steady loss and replenishment during skin homeostasis. Merkel cells develop concomitant with dermal appendages along the trunk and loss of Ectodysplasin signaling, which prevents dermal appendage formation, reduces Merkel cell density by affecting cell differentiation. By contrast, altering dermal appendage morphology changes the distribution, but not density, of Merkel cells. Overall, our studies provide insights into touch system maturation during skin organogenesis and establish zebrafish as an experimentally accessible in vivo model for the study of Merkel cell biology.

## Editor's evaluation

The authors describe and characterize the touch system in zebrafish as a new model to study Merkel cell development and maintenance. The study demonstrates that the zebrafish touch system shares many characteristics with its mammalian counterpart, including developmental origin, innervation, and molecular characteristics while allowing in vivo analysis of specification, development, and maintenance. This study is the foundation for future detailed cellular and molecular analyses of the touch sensory system and will be of interest to developmental biologists and neuroscientists studying stem cells, regeneration, and aging.

## Introduction

Skin functions as our primary interface with the physical environment and can distinguish a range of tactile inputs with exquisite acuity. As the skin undergoes organogenesis, the epidermis transforms from a simple, uniform epithelium into a complex, diverse tissue. During these dramatic changes, the

**\*For correspondence:**
rasmuss@uw.edu

†These authors contributed equally to this work

**Competing interest:** The authors declare that no competing interests exist.

skin develops regionally specialized sensory structures and becomes innervated by specific types of somatosensory neurites (reviewed by *Jenkins and Lumpkin, 2017*). Interactions between somatosensory neurites and cutaneous cell types regulate diverse tactile responses (reviewed by *Handler and Ginty, 2021*). Altered tactile sensitivity during early mammalian development has been associated with neurodevelopmental disorders (reviewed by *Orefice, 2020*), underscoring the importance of understanding the cellular and molecular basis of touch system development and function.

Merkel cells (MCs), a specialized mechanosensory cell type found in the vertebrate epidermis (reviewed by *Hartschuh et al., 1986*), densely populate many highly sensitive regions of skin (*Lacour et al., 1991*). MCs have several defining cellular characteristics that distinguish them from other epidermal cell types: they are relatively small, extend actin-rich microvilli, contain cytoplasmic granules reminiscent of synaptic vesicles, and form contacts with somatosensory axons (*Hartschuh and Weihe, 1980*; *Mihara et al., 1979*; *Smith, 1977*; *Toyoshima et al., 1998*). In mammals, a subset of cutaneous somatosensory axons known as Aß slowly adapting type I low-threshold mechanoreceptors (SAI-LTMRs) innervate MCs, forming the MC-neurite complex. MCs detect mechanical inputs via the cation channel Piezo2 (*Ikeda et al., 2014*; *Maksimovic et al., 2014*; *Woo et al., 2014*) and play an active role in touch sensation by releasing neurotransmitters to activate neighboring neurites (*Chang et al., 2016*; *Chang and Gu, 2020*; *Hoffman et al., 2018*). Genetic ablation of rodent MCs indicates they are required for specific aspects of touch system function, including promoting the static phase of the slowly adapting response of Aß SAI-LTMRs and sensory tasks such as texture discrimination (*Maricich et al., 2012*; *Maricich et al., 2009*).

Molecular control of MC development has primarily been studied in rodent hairy skin (reviewed by *Oss-Ronen and Cohen, 2021*). While this system has been useful for understanding many aspects of MC development and function, the rodent system also has several significant limitations that warrant additional models to improve the understanding of MCs. First, vertebrates have diverse types of skin, and MCs are found in both hairy and glabrous (non-hairy) skin, as well as mucocutaneous regions such as the gingiva and palate (*Hashimoto, 1972*; *Lacour et al., 1991*; *Moayedi et al., 2021*). Importantly, MC populations within different skin compartments share similar transcriptional profiles (*Nguyen et al., 2019*). Thus, the establishment of complementary genetic systems in different types of skin could help reveal both shared and divergent principles of MC development. Second, because in utero development of mammalian skin limits access to the developing touch system—combined with technical limitations of imaging intact mammalian skin—the dynamics of MC development and innervation remain essentially unknown. Third, unbiased screens for regulators of MC development would be difficult or impractical in rodents due to the prohibitive cost of animal housing and difficulty of visualizing MCs in situ.

Anamniote model systems, such as the genetically tractable zebrafish, provide the potential to overcome these limitations. Interestingly, despite the different tactile environments encountered by terrestrial and aquatic vertebrates, MCs have been described by transmission electron microscopy (TEM) in a wide variety of anamniotes, including teleost (ray-finned) fish, lungfish, and lamprey (*Fox et al., 1980*; *Lane and Whitear, 1977*; *Whitear, 1989*; *Whitear and Lane, 1981*). Here, we identify and characterize a population of zebrafish epidermal cells that we propose are bona fide MCs. Our studies establish the zebrafish as a promising new model to investigate the developmental and cellular biology of MCs.

## Results

### Ultrastructural identification of presumptive MCs in the adult epidermis

Given the presence of cells with the ultrastructural characteristics of MCs in several teleosts (*Lane and Whitear, 1977*; *Whitear, 1989*), we reasoned that the zebrafish epidermis may contain similar cells. *Whitear, 1989* defined five ultrastructural criteria for the identification of vertebrate MCs: (1) a relatively small volume of cytoplasm; (2) an association with a nerve fiber; (3) the presence of cytoplasmic granules; (4) desmosomal attachments to neighboring cells; and (5) peripheral microvilli.

We previously demonstrated that somatosensory axons densely innervate the epidermis above scales (*Rasmussen et al., 2018*), dermal appendages that cover the adult zebrafish trunk (*Figure 1A*). By TEM, we found that many of the axon endings in the scale epidermis arborize between keratinocyte membranes (*Rasmussen et al., 2018*). Interestingly, however, we identified additional axon-associated

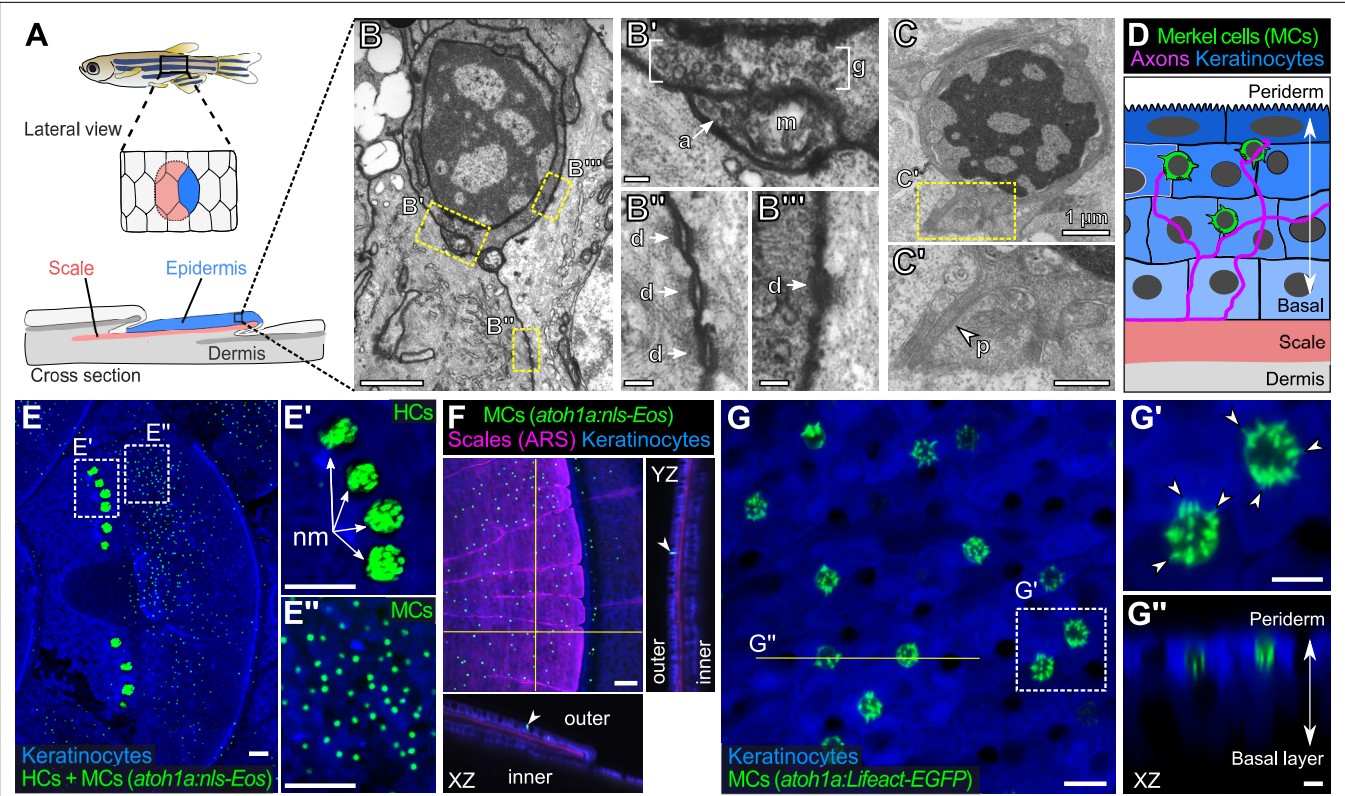

**Figure 1.** The adult scale epidermis contains *atoh1a+* Merkel cells (MCs). (**A**) Illustration of the adult zebrafish trunk anatomy showing the organization of epidermis, scales, and dermis. Scales are flat bony discs arranged in an overlapping, imbricated pattern and coated on their external surface by epidermis. (**B**) Transmission electron microscopy (TEM) of a presumptive MC from the scale epidermis. Dotted boxes indicate regions of magnification in **B′–B‴**. (**B′**) Magnification of B showing cytoplasmic granules (g, brackets) juxtaposed to a putative axon (a) contact containing a mitochondrion (m). (**B″ and B‴**) Magnifications of B showing desmosomal-like (d, arrows) attachments between keratinocytes (**B″**) and between a presumptive MC and keratinocyte (**B‴**). (**C and C′**) TEM of a presumptive MC from the scale epidermis showing a microvillar process (p, arrowhead). (**D**) Illustration of a cross section of the scale epidermis based on TEM observations. Periderm cells (superficial epidermis; dark blue) are located in the uppermost epidermal stratum, and basal keratinocytes (light blue) are located in the lowermost epidermal stratum. MCs containing cytoplasmic granules, extending microvillar processes, and contacting axons localize between keratinocytes. (**E**) Lateral confocal micrograph of the trunk epidermis in an adult expressing reporters for keratinocytes (*Tg(actb2:LOXP-BFP-LOXP-DsRed)*) and *atoh1a*-expressing cells (*Tg(atoh1a:nls-Eos)*). Dotted boxes indicate areas of magnification in **E′ and E″**. (**E′**) Magnification of E showing *atoh1a+* hair cells (HCs) and progenitors within neuromasts (nm) of the posterior lateral line. (**E″**) Magnification of E showing *atoh1a+* MCs scattered throughout the scale epidermis. (**F**) Lateral and reconstructed cross sectional confocal micrographs of the trunk in an adult expressing reporters for keratinocytes (*Tg(actb2:LOXP-BFP-LOXP-DsRed)*) and *atoh1a*-expressing cells (*Tg(atoh1a:nls-Eos)*) and stained with Alizarin Red S (ARS) to label the mineralized scale matrix. Note that *atoh1a+* MCs localize to the epidermis above scales (arrowhead). (**G**) Lateral confocal micrograph of the scale epidermis in an adult expressing reporters for keratinocytes (*Tg(krt4:DsRed)*) and F-actin within *atoh1a+* MCs (*Tg(atoh1a:Lifeact-EGFP)*). Note that all *atoh1a+* MCs extend multiple microvilli. (**G′**) Magnification of G with arrowheads indicating individual microvillar processes on the surface of MCs. (**G″**) Reconstructed cross section along the yellow line in G. MCs localize to the upper epidermal strata as diagrammed in D. Note that *Tg(krt4:DsRed)* (blue) preferentially labels keratinocytes in the upper epidermal strata, but not in the basal cell layer. Scale bars: 1 μm (**B and C**), 0.1 μm (**B′–B‴**), 0.5 μm (**C′**), 50 μm (**E–E″ and F**), 10 μm (**G**), and 5 μm (**G′ and G″**).

The online version of this article includes the following source data and figure supplement(s) for figure 1:

**Figure supplement 1.** TEM characterization of presumptive MC and adjacent keratinocytes.

**Figure supplement 2.** Characterization of *atoh1a* reporter transgenes in larvae.

**Figure supplement 3.** MCs in the adult epidermis express Sox2.

**Figure supplement 3—source data 1.** Datasheet for *Figure 1—figure supplement 3*.

epidermal cells that were distinct from the large, cuboidal keratinocytes that comprise most of the epidermis based on several characteristics. The cells appeared relatively small and spherical with a low cytoplasmic-to-nuclear ratio compared to neighboring keratinocytes (*Figure 1B and C*; *Figure 1—figure supplement 1*), contained cytoplasmic vesicles that in some instances localized adjacent to axon contacts (*Figure 1B′*), and formed desmosomal-like attachments with neighboring keratinocytes

(*Figure 1B" and B'''*). Furthermore, the cells extended spike-like microvillar processes that contacted adjacent cells (*Figure 1C and C'*). Thus, based on established TEM criteria, we identified presumptive MCs in the adult scale epidermis.

## *atoh1a* reporters label MCs in the adult epidermis

To date, a lack of genetically encoded reagents has hindered in-depth study of anamniote MCs. Since the TEM studies of Whitear and colleagues decades ago, molecular markers have been identified that distinguish mammalian MCs from other epidermal cells. For example, keratins, most notably keratin 8 and keratin 20, have been used extensively as markers of mammalian MCs (*Moll et al., 1995*; *Moll et al., 1984*). However, teleost keratins have undergone extensive gene loss and duplication and are not orthologous to mammalian keratin genes (*Ho et al., 2022*). Thus, we considered alternative molecular markers to label zebrafish MCs. Expression of Atoh1 uniquely identifies MCs in rodent skin and is necessary and sufficient for MC development (*Morrison et al., 2009*; *Ostrowski et al., 2015*; *Van Keymeulen et al., 2009*). The zebrafish genome contains three genes (*atoh1a*, *atoh1b*, and *atoh1c*) encoding Atoh1 homologs (*Chaplin et al., 2010*; *Kani et al., 2010*). To determine if the adult epidermis contained cells expressing an Atoh1 homolog, we focused on characterizing the expression pattern of *atoh1a* due to the availability of an enhancer trap line that expresses a nuclear localized version of the photoconvertible fluorescent protein Eos (nls-Eos) from the endogenous *atoh1a* locus (*Tg(atoh1a:nls-Eos)*; *Pickett et al., 2018*). Confocal imaging of the adult trunk revealed that *Tg(atoh1a:nls-Eos)* labeled hair cells of the posterior lateral line, which formed tight clusters within neuromasts in interscale regions (*Figure 1E and E'*). In addition to *atoh1a+* cells of the lateral line, we identified a second, spatially distinct population of *atoh1a+* cells dispersed across the scale surface (*Figure 1E, E" and F*). Reconstructed cross sections showed that this population of *atoh1a+* cells resided within the epidermis above scales (*Figure 1F*), in a similar axial position to the cells we identified by TEM.

The numerous, actin-rich microvilli that emanate from the MC surface morphologically distinguish them from other epidermal cells (*Lane and Whitear, 1977*; *Toyoshima et al., 1998*; *Yamashita et al., 1993*). To determine whether the dispersed epidermal *atoh1a+* cells extended microvilli, we created an *atoh1a* enhancer trap line that expresses Lifeact-EGFP, a reporter for filamentous actin (*Riedl et al., 2008*). Similar to *Tg(atoh1a:nls-Eos)*, *Tg(atoh1a:Lifeact-EGFP)* labeled hair cells of the lateral line and inner ear in larvae (*Figure 1—figure supplement 2*). *atoh1a+* cells were notably absent from regions above the larval eye, yolk sac, or caudal fin (*Figure 1—figure supplement 2*), where neuroepithelial cells (NECs), a morphologically distinct population of sensory cells, have been described in larval skin (*Coccimiglio and Jonz, 2012*). Confocal microscopy of the scale epidermis in *Tg(atoh1a:Lifeact-EGFP)* adults revealed actin-rich microvilli densely decorating *atoh1a+* cells in close proximity to neighboring keratinocytes (*Figure 1G*), further suggesting that the epidermal *atoh1a+* cell population in the trunk shared key characteristics with the candidate MCs identified by TEM. Immunostaining for Sox2, a transcription factor required for murine MC maturation (*Bardot et al., 2013*; *Perdigoto et al., 2014*), demonstrated that the epidermal *atoh1a+* cells expressed Sox2 (*Figure 1—figure supplement 3*). Together, these results define molecular and cellular properties of a previously uncharacterized epidermal cell population in zebrafish and identify genetic reagents for the study of this cell type. Anticipating the conclusion of our analysis below, we shall hereafter refer to the epidermal *atoh1a+* cells as MCs, with the majority of the analyses completed on trunk MCs unless stated otherwise.

## Somatosensory axons innervate zebrafish MCs, which display neurosecretory and mechanosensory characteristics

We next sought to determine whether zebrafish MCs displayed other key characteristics of MCs defined in mammals, including innervation by somatosensory axons and expression of neurosecretory and mechanosensory machinery.

Our ultrastructural observations suggested that cutaneous axons innervate MCs (*Figure 1B*). Staining scales with zn-12, a monoclonal antibody that labels several types of peripheral axons (*Metcalfe et al., 1990*), revealed that >90% of MCs were tightly associated with axons (*Figure 2A and C*). To determine the type of axon(s) innervating MCs, we examined expression of genetically encoded somatosensory axon reporters that we previously characterized in adult scales (*Rasmussen et al., 2018*). Analysis of reporters for three somatosensory neuron-expressed genes (*p2rx3a*, *p2rx3b*,

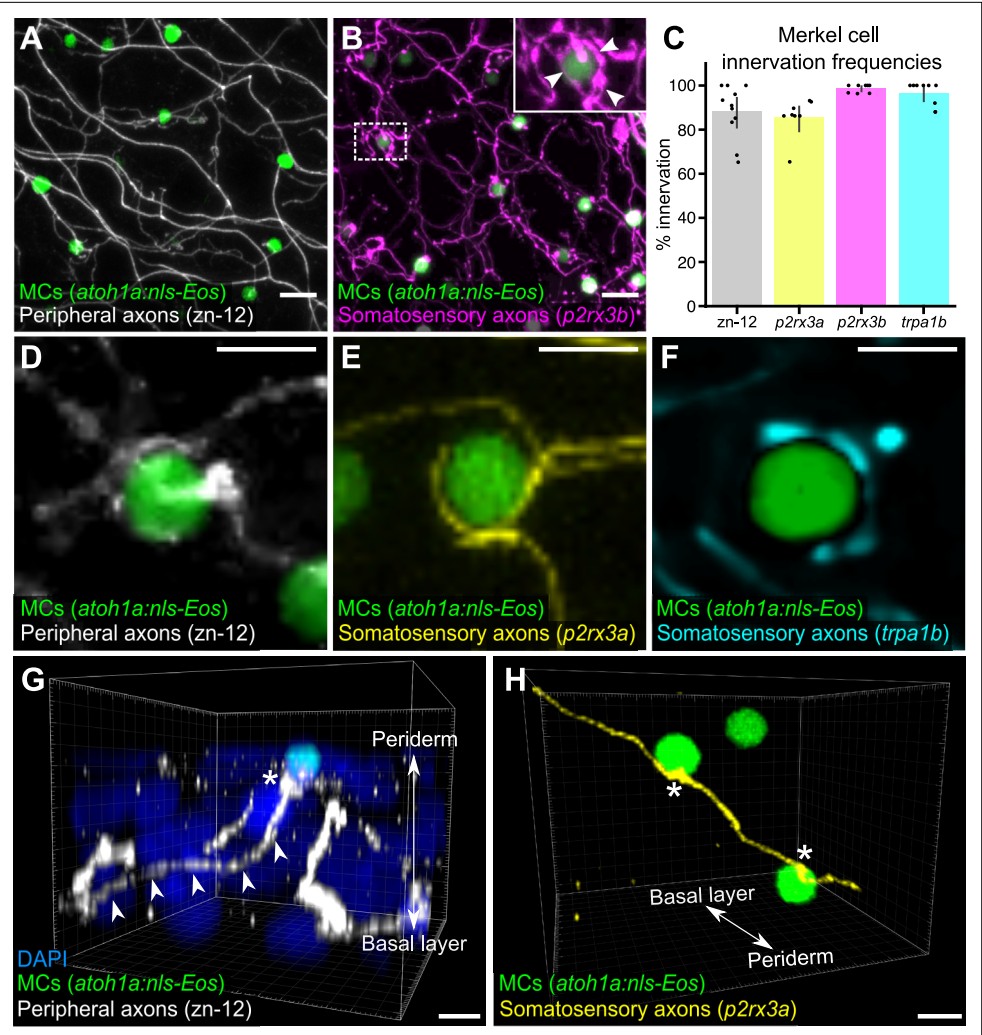

**Figure 2.** Somatosensory axons innervate Merkel cells (MCs) in the adult epidermis. (**A**) Lateral confocal micrograph of the scale epidermis from an adult expressing an MC reporter immunostained for peripheral axons (zn-12). (**B**) Lateral confocal micrograph of the scale epidermis showing that somatosensory peripheral axons (*Tg(p2rx3b:EGFP)*) innervate MCs. Inset of dotted region shows axonal varicosities adjacent to an MC (arrowheads). (**C**) Quantification of MC innervation in the scale epidermis (17–30 mm standard length [SL]). Each dot represents measurements from an individual scale. Innervation frequencies: zn-12, 91% (284/311 cells; N=3 adults); *Tg(p2rx3a>mCherry)*, 86% (196/228 cells; N=4 adults); *Tg(p2rx3b:EGFP)*, 99% (225/228 cells; N=4 adults); *Tg(trpa1b:EGFP)*, 96% (217/225 cells; N=9 adults). Error bars represent 95% CIs. (**D–F**) High-magnification confocal micrographs showing examples of somatosensory axons forming extended, ring-like contacts with MCs within the scale epidermis. (**G**) Three-dimensional (3D) reconstruction of an axon (zn-12 immunostaining, arrowheads) forming a bouton-like ending (asterisk) that terminates in close proximity to an MC. DAPI staining labels epidermal nuclei. (**H**) 3D reconstruction of a single somatosensory axon (*Tg(p2rx3a>mCherry)*) that forms en passant-like contacts (asterisks) with multiple MCs. Scale bars: 10 μm (**A and B**), 5 μm (**D–H**).

The online version of this article includes the following source data for figure 2:

**Source code 1.** ImageJ macro used for *Figure 2C*.

**Source data 1.** Datasheet for *Figure 2C*.

and *trpa1b*; *Kucenas et al., 2006*; *Palanca et al., 2013*; *Pan et al., 2012*) demonstrated that somatosensory axons innervated up to 99% of MCs (*Figure 2B and C*). Consistent with ultrastructural analyses of MCs in the skin of other teleosts (*Whitear, 1989*), some axons formed ring-like structures that wrapped around MCs with MC-axon contacts containing varicosities or swellings (*Figure 2B*, inset and *Figure 2D-F*; *Video 1*). Additionally, we observed examples of axons forming both bouton- and

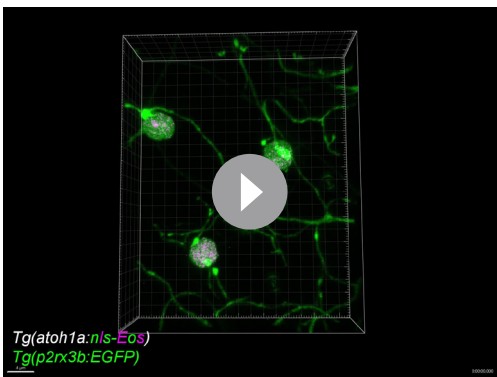

**Video 1.** Three-dimensional (3D) reconstruction of somatosensory axon and MC interactions. 3D rotation of somatosensory axons (green) and photoconverted MCs (green and magenta) in the adult scale epidermis. Arrows indicate axonal varicosities in close proximity to MCs. Scale bar: 4 μm.

https://elifesciences.org/articles/85800/figures#video1

en passant-like contacts with MCs (*Figure 2G and H*; *Figure 1D*).

Based on our observation that MCs contained cytoplasmic granules (*Figure 1B*), we postulated that they would display neurosecretory characteristics. We began by staining scales with an antibody against synaptic vesicle glycoprotein 2 (SV2), which labels secretory vesicle membranes in zebrafish and mammalian neuronal and endocrine cells (*Buckley and Kelly, 1985*; *Jonz and Nurse, 2003*). Essentially all MCs contained SV2-positive structures (*Figure 3A*), suggesting they express neurosecretory machinery that may contain neurotransmitter(s). Indeed, immunostaining revealed that MCs expressed serotonin (5-hydroxytryptamine; 5-HT) (*Figure 3B*), similar to mammalian MCs (*Chang et al., 2016*; *English et al., 1992*; *García-Caballero et al., 1989*) and zebrafish NECs (*Coccimiglio and Jonz, 2012*). Both 5-HT and SV2 appeared in a speckled pattern within MCs (*Figure 3A and B*), consistent with a vesicular localization.

Do zebrafish MCs exhibit properties consistent with mechanosensory function? To address this question, we began by staining scales with AM1-43, an activity-dependent fluorescent styryl dye that labels a variety of sensory cells, including mammalian MCs (*Meyers et al., 2003*) and hair cells of the zebrafish lateral line (*Corey et al., 2004*). Following a short preincubation, AM1-43 robustly stained MC membranes and punctate structures reminiscent of vesicular compartments (*Figure 3C*), suggestive of ion channel expression in MCs (*Meyers et al., 2003*). Mammalian MCs express the mechanically activated cation channel Piezo2, which is required for MC mechanosensory responses (*Ikeda et al., 2014*; *Maksimovic et al., 2014*; *Woo et al., 2014*). Hybridization chain reaction (HCR) with an antisense probeset against *piezo2* labeled MCs in adult scales (*Figure 3D*). We confirmed this staining pattern by performing fluorescent in situ hybridization (FISH) with a previously described *piezo2* probe (*Faucherre et al., 2013*; *Figure 3—figure supplement 1*). Together, these data suggest that somatosensory peripheral axons innervate adult MCs, which possess neurosecretory and mechanosensory properties.

## MCs arise from basal keratinocyte precursors

What are the precursors of MCs in zebrafish? Analysis of MC progenitors have come to conflicting results in avians and rodents: quail-chick chimeras suggest a neural crest origin for avian MC (*Grim and Halata, 2000*), whereas Cre-based lineage tracing studies in mouse demonstrate an epidermal origin (*Morrison et al., 2009*; *Van Keymeulen et al., 2009*).

To investigate a possible neural crest origin, we crossed a Cre driver expressed in neural crest progenitors (*Tg(sox10:Cre)*; *Kague et al., 2012*) to a reporter transgene that stably expresses DsRed upon Cre-mediated recombination from a quasi-ubiquitous promoter (*Tg(actb2:LOXP-BFP-LOXP-DsRed)*; *Kobayashi et al., 2014*; *Figure 4A*). DsRed+ neural crest-derived cell types, such as Schwann cells, appeared along scales, indicative of successful recombination (*Figure 4B*). However, we observed <0.5% co-labeling between the neural crest lineage trace and an MC reporter (*Figure 4B' and F*). Based on these results, we concluded that zebrafish MCs likely derive from a non-neural crest lineage.

To investigate a possible epidermal origin, we considered basal keratinocytes, an epidermal-resident stem cell population, the most likely candidate progenitors. To follow this lineage, we engineered a transgene to express a tamoxifen-inducible Cre recombinase from regulatory sequences of ΔNp63 (*TgBAC(ΔNp63:Cre-ERT2)*), a basal keratinocyte marker (*Bakkers et al., 2002*; *Lee and Kimelman, 2002*). We crossed this transgene to the Cre reporter transgene and treated embryos with 4-hydroxytamoxifen (4-OHT) at 1 day post-fertilization (dpf) to induce Cre-ERT2 activity, which

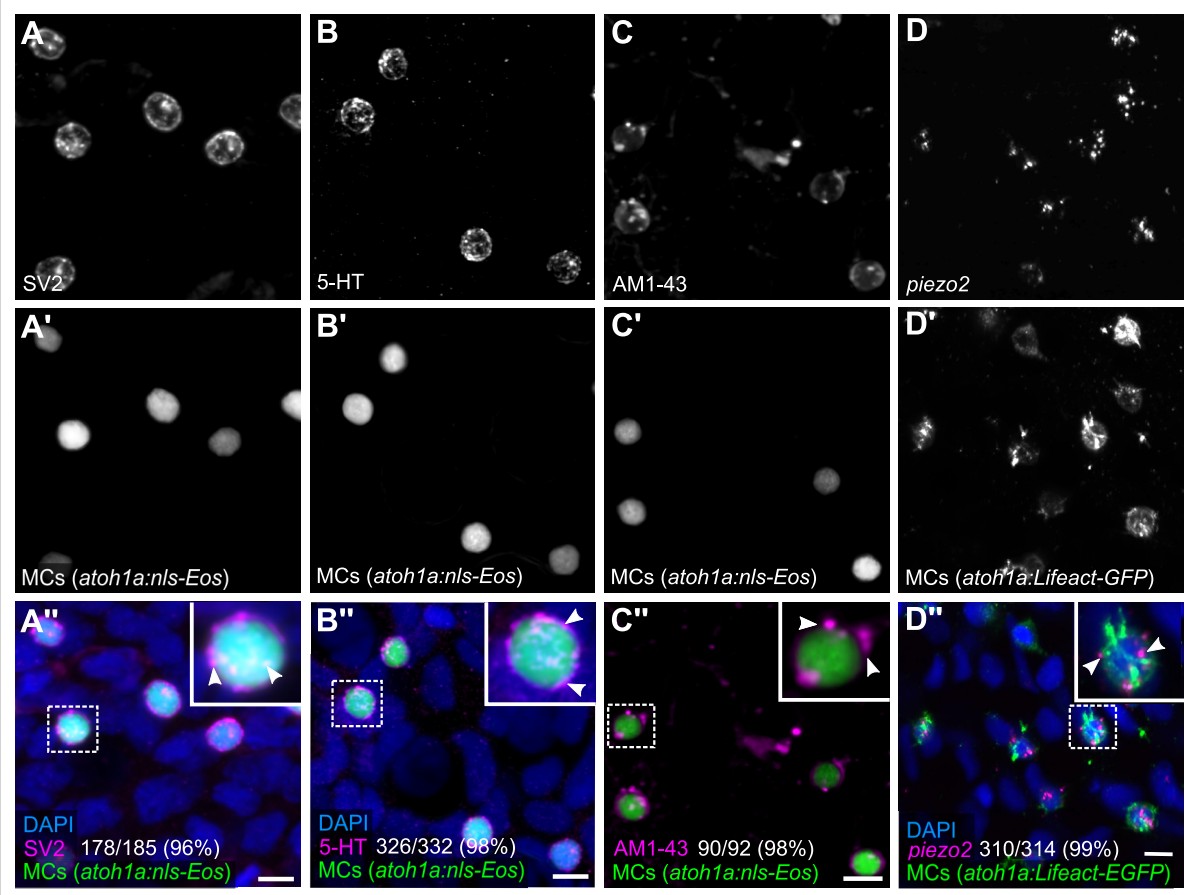

**Figure 3.** Merkel cells (MCs) in the adult epidermis express neurosecretory and mechanosensory machinery. (**A and B**) Anti-SV2 (**A–A"**) or anti-5-hydroxytryptamine (5-HT) (**B–B"**) immunostaining of the scale epidermis from an adult expressing an MC reporter. Insets of dotted regions show the punctate localization of SV2 and 5-HT staining in MCs (arrowheads), consistent with a vesicular localization. 96% of MCs (178/185) were SV2+. 98% of MCs (326/332) were 5-HT+. Cells analyzed from n=3 scales from N=2 adults (25–27 mm standard length [SL]). DAPI labels epidermal nuclei. (**C**) Scale epidermis from an adult expressing an MC reporter stained with AM1-43. 98% of MCs (90/92) were AM1-43+. Cells analyzed from n=6 scales from N=2 adults. Inset of dotted region shows puncta within an MC labeled by AM1-43 (arrowheads). AM1-43 has been reported to stain neurites innervating MCs in murine whisker vibrissae (*Meyers et al., 2003*). However, our AM1-43 staining regimen did not strongly label cutaneous axons, although we cannot exclude low levels of staining. (**D**) Scale epidermis from an adult expressing an MC reporter stained with hybridization chain reaction (HCR) probes against *piezo2* and an anti-GFP antibody. 99% of MCs (310/314) were *piezo2*+. Cells analyzed from n=7 scales from N=2 adults. Arrowheads indicate examples of positive staining within an MC. Scale bars: 5μm.

The online version of this article includes the following figure supplement(s) for figure 3:

**Figure supplement 1.** MCs in the adult epidermis express *piezo2*.

resulted in permanent DsRed expression in basal keratinocytes and their derivatives (*Figure 4C and D*; *Figure 4—figure supplement 1*). After raising 4-OHT-treated animals to adulthood, we observed variable (2–81%) co-labeling between the basal keratinocyte lineage trace and MC reporters (*Figure 4D' and F*). We note that our lineage tracing strategy did not label all basal keratinocytes (*Figure 4D*; *Figure 4—figure supplement 1*), suggestive of incomplete Cre-ERT2 induction and/or transgene recombination. Consistent with the latter possibility, a recent analysis demonstrated *Tg(actb2:LOXP-BFP-LOXP-DsRed)* has a low recombination efficiency compared to other Cre reporter transgenes (*Lalonde et al., 2022*). To estimate the local recombination efficiency in imaged regions, we thresholded the DsRed channel and calculated the fraction of skin cells labeled (*Figure 4E*). Importantly, the proportion of MCs labeled by the basal keratinocyte lineage trace was not significantly different from the local recombination efficiency (*Figure 4G–H*). These observations support a basal keratinocyte origin of most or all zebrafish MCs.

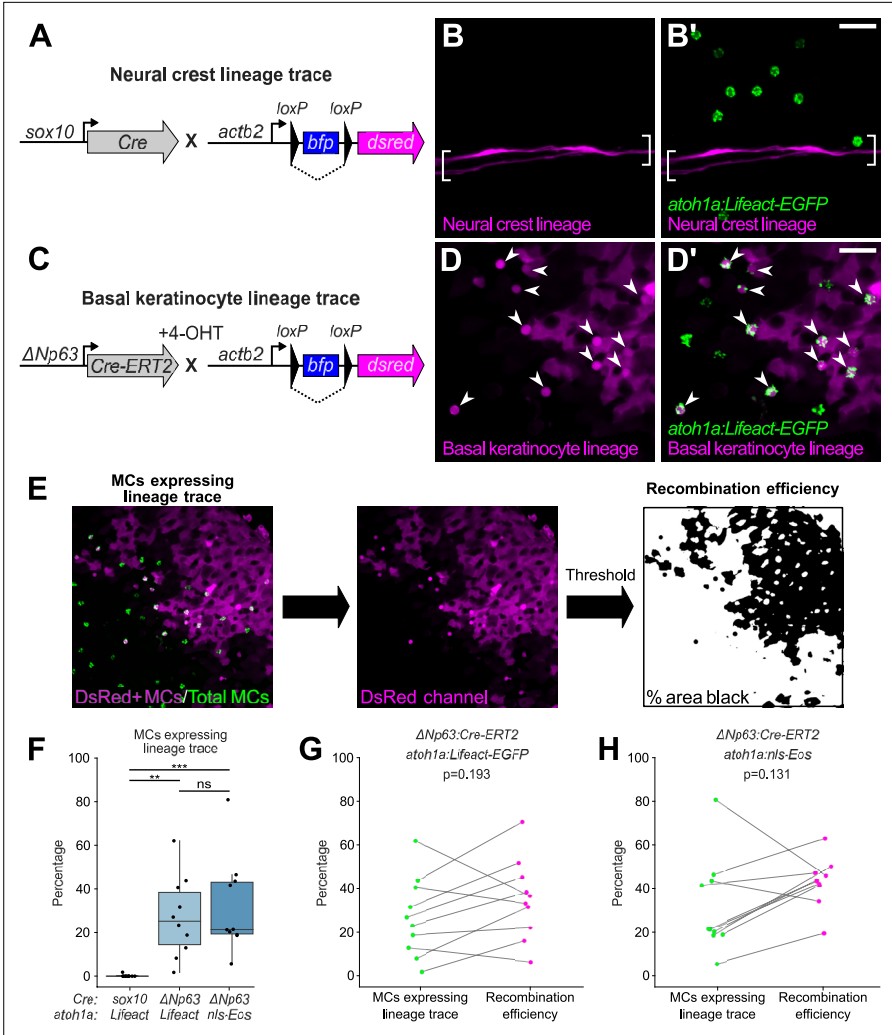

**Figure 4.** Merkel cells (MCs) derive from the basal keratinocyte lineage. (**A**) Schematic of Cre-based neural crest lineage tracing strategy. (**B**) Confocal micrograph of the scale epidermis in an adult expressing neural crest lineage (*Tg(sox10:Cre); Tg(actb2:LOXP-BFP-LOXP-DsRed)*) and MC (*Tg(atoh1a:Lifeact-EGFP)*) reporters. Brackets denote Schwann cells associated with a nerve along a scale radius. (**C**) Schematic of Cre-based basal keratinocyte lineage tracing strategy. (**D**) Confocal micrograph of the scale epidermis in an adult expressing basal keratinocyte lineage (*TgBAC(ΔNp63:Cre-ERT2); Tg(actb2:LOXP-BFP-LOXP-DsRed)*) and MC (*Tg(atoh1a:Lifeact-EGFP)*) reporters, which was treated with 4-hydroxytamoxifen (4-OHT) at 1 day post-fertilization (dpf). Arrowheads indicate MCs labeled by the basal keratinocyte lineage reporter. Note that recombination is not complete, possibly explaining why not all MCs express the lineage reporter. (**E**) Workflow to calculate percentage of MCs expressing lineage reporter and percentage of total cells expressing lineage reporter. (**F**) Boxplots of the percentage of MCs expressing the lineage tracing reporters diagrammed in panels A and C. Each dot represents an individual scale. Overall percentage of MCs expressing lineage trace reporters: *sox10/Lifeact*, 0.3% (1/323 cells; N=6 adults, 27.5–31 mm standard length [SL]); *ΔNp63/Lifeact*, 29.7% (299/1005 cells; N=6 adults, 21–26 mm SL); *ΔNp63/nls-Eos*, 32.3% (386/1195 cells; N=4 adults, 20–30 mm SL). A one-way ANOVA (F=12.06; p<0.001) with Tukey's post-hoc honestly significant difference (HSD) test was used to compare groups. \*\*, p<0.01; \*\*\*, p<0.001. (**G and H**) Paired dot plots of the percentage of MCs expressing the indicated *atoh1a* reporter and the basal keratinocyte lineage reporter compared to the percentage of all cells in the field of view expressing DsRed. Statistical analyses were performed using the Wilcoxon test. Scale bars: 20 µm.

The online version of this article includes the following source data and figure supplement(s) for figure 4:

**Source data 1.** Datasheet for *Figure 4F*.

**Source data 2.** Datasheet for *Figure 4G and H*.

**Figure supplement 1.** Validation of basal keratinocyte lineage tracing strategy.

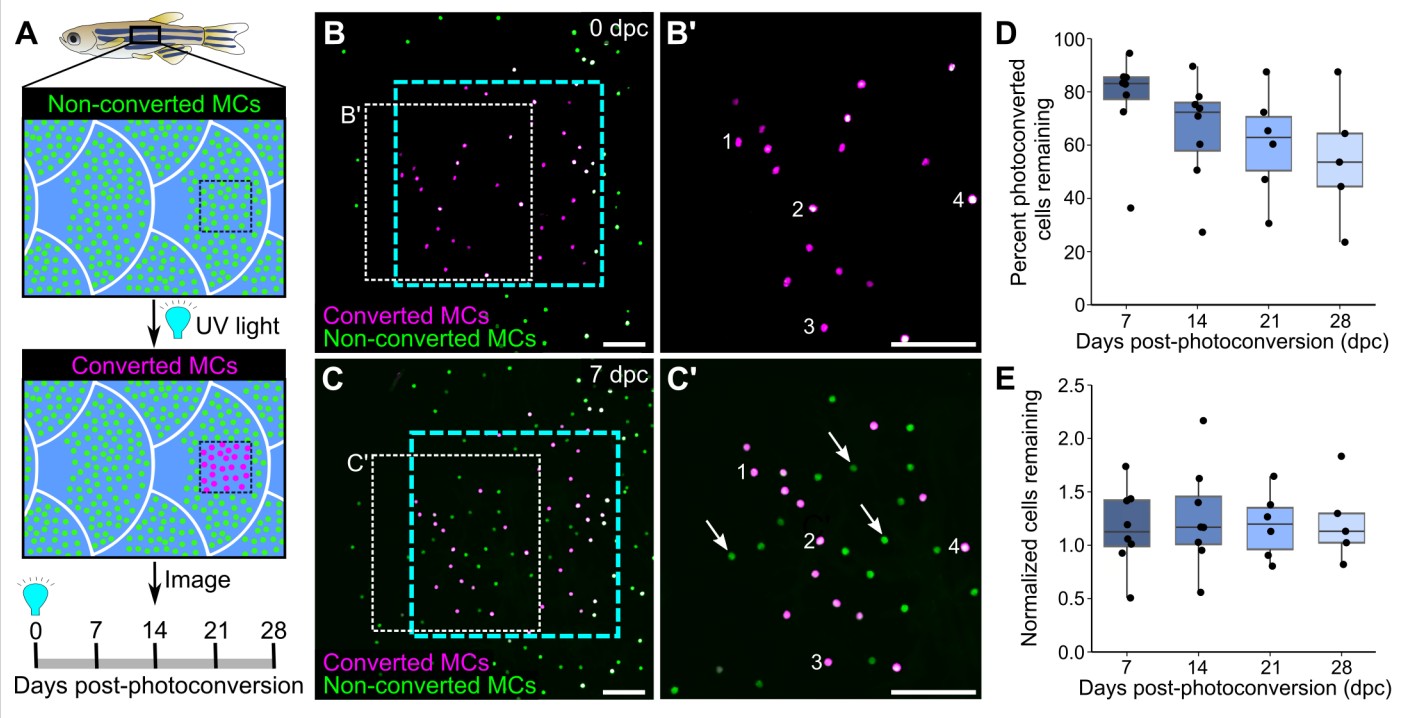

**Figure 5.** Homeostatic replacement of Merkel cells (MCs) in the adult epidermis. (**A**) Illustration of the photoconversion experiment showing the epidermis (blue), non-converted MCs (green), and converted MCs (magenta) after exposure of a region of the scale epidermis to UV light. (**B and C**) Representative images of MCs labeled by *Tg(atoh1a:nls-Eos)* at 0 (**B**) or 7 (**C**) days post-conversion (dpc) from a single adult. Cyan dotted box indicates the photoconverted region. White dotted box indicates the area magnified in B' and C'. (**B' and C'**) Numbers label examples of individual cells present at 0 and 7 dpc. Arrows indicate examples of newly added cells, which appear green due to the presence of non-converted nls-Eos (green) and absence of converted nls-Eos (magenta). (**D and E**) Boxplots of the percentage of photoconverted MCs remaining compared to 0 dpc (**D**) and the total number of MCs (converted+non-converted) present at each day compared to 0 dpc (**E**). Each dot represents an individual fish. N=5–8 fish (24–32 mm standard length [SL]). Scale bars: 50 µm.

The online version of this article includes the following source data for figure 5:

**Source data 1.** Datasheet for *Figure 5D and E*.

## MCs continuously turn over in adult skin

The longevity and turnover of murine MCs are controversial. Several studies concluded that MC numbers fluctuate with hair cycle stages (*Marshall et al., 2016*; *Moll et al., 1996*; *Nakafusa et al., 2006*), while another found no correlation (*Wright et al., 2017*). To determine the turnover rate of zebrafish MCs, we photoconverted small regions of the scale epidermis in *Tg(atoh1a:nls-Eos)* adults and tracked individual cells over time. Exposure to UV light irreversibly photoconverts nls-Eos, allowing us to distinguish pre-existing cells (containing photoconverted nls-Eos) from newly added cells (without photoconverted nls-Eos; *Figure 5A–C*). By longitudinally tracking individual fish over the course of 28 days, we found a decrease of ~15% of the photoconverted MCs every 7 days (*Figure 5D*). In addition to the gradual loss of MCs over time, we noted a steady addition of new MCs (Figure 5C'), resulting in a nearly constant total cell number (*Figure 5E*). Thus, MCs undergo constant cell loss and renewal in adult skin, albeit at a slower rate than *atoh1a*-expressing hair cells of the lateral line (*Cruz et al., 2015*).

## MCs are widely distributed across the body, in compartment-specific patterns

MCs localize to specific regions of mammalian skin, such as in crescent-shaped touch domes adjacent to hair follicles in hairy skin and at the bottom of rete ridges in glabrous skin (*Boot et al., 1992*; *Fradette et al., 1995*; *Iggo and Muir, 1969*; *Lacour et al., 1991*). To determine the distribution pattern of zebrafish MCs, we used confocal microscopy to survey multiple regions of the adult skin.

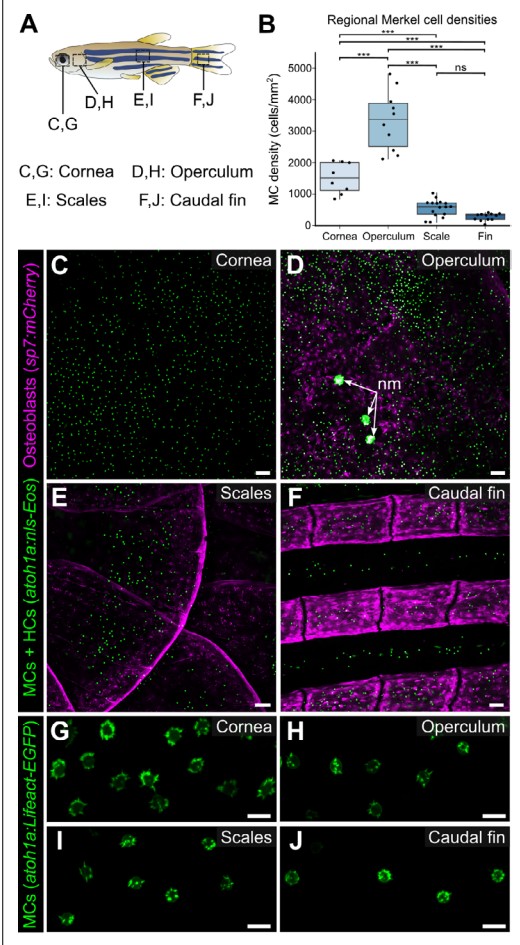

**Figure 6.** Merkel cells (MCs) are widely distributed across the skin, in compartment-specific patterns. (**A**) Illustration indicating the epidermal regions imaged in adult zebrafish. (**B**) Quantification of MC densities in the specified regions. Each dot represents an individual fish (N=8–18, 20–29.5 mm standard length [SL]). *** indicates p<0.001 using a one-way ANOVA (F=83.94; p<0.001) with post-hoc Tukey's HSD test. (**C–J**) Lateral confocal micrographs of MCs in the different skin regions from animals expressing the indicated reporters. The regions imaged are indicated in **A**. Note that MCs expressing *Tg(atoh1a:Lifeact-EGFP)* have a similar morphology across skin compartments (**G–J**). nm, neuromasts of the posterior lateral line. Scale bars: 50 μm (**C–F**) and 10 μm (**G–J**).

The online version of this article includes the following source data for figure 6:

**Source data 1.** Datasheet for *Figure 6B*.

In addition to the MCs found on the trunk, MCs appeared in the epidermis above the eyes, gill covers (opercula), and fins (*Figure 6A–F*). While MC morphology was similar across the skin compartments (*Figure 6G–J*), MC densities and spatial distributions varied across skin compartments (*Figure 6B*). For example, MCs were distributed relatively uniformly across the cornea, although this spatial pattern was not specifically quantified (*Figure 6C*). By contrast, in the caudal fin, MCs localized to the epidermis above bony rays and in the medial regions of the interray epidermis between bony rays (*Figure 6F*). Along the trunk, MCs appeared in patches, similar to the pattern of dermal scales beneath the epidermis (*Figure 6E*). Altogether, our results demonstrate that MCs are widely distributed across the adult zebrafish skin and localize in specific patterns in each skin compartment.

## Trunk MCs develop concomitant with dermal appendage morphogenesis

To examine the mechanisms that generate a compartment-specific MC pattern, we focused on the trunk skin because of its molecular and cellular similarities to murine hairy skin (*Aman et al., 2018*; *Harris et al., 2008*). Both during ontogeny and at post-embryonic stages, murine MCs associate with primary (guard) hairs, a subclass of dermal appendages (*Jenkins et al., 2019*; *Nguyen et al., 2018*; *Perdigoto et al., 2016*). Based on these studies in mice, and our previous work showing that epidermal diversification and somatosensory remodeling coincides with scale development in zebrafish (*Rasmussen et al., 2018*), we postulated that MCs would appear during squamation (scale formation).

Zebrafish post-larval development is staged by standard length (SL) in millimeters (mm; *Parichy et al., 2009*). Squamation begins at ~9 mm SL (*Figure 7A*; *Aman et al., 2018*; *Harris et al., 2008*; *Sire et al., 1997a*). Using reporters that label MCs and scale-forming osteoblasts, we rarely observed MCs in the epidermis prior to 8 mm SL (*Figure 7B and F*). Between 8 and 10 mm SL, MCs appeared at a low density along the trunk (*Figure 7C and F*). MC density rapidly increased from 10 to 15 mm SL, a period of active scale growth (*Figure 7D, E and F*). The density and number of MCs positively correlated with scale area (*Figure 7G and H*), although this trend was less pronounced at stages less than 10 mm SL (*Figure 7—figure supplement 1*). These data indicate that MC development coincides with dermal appendage growth along the trunk.

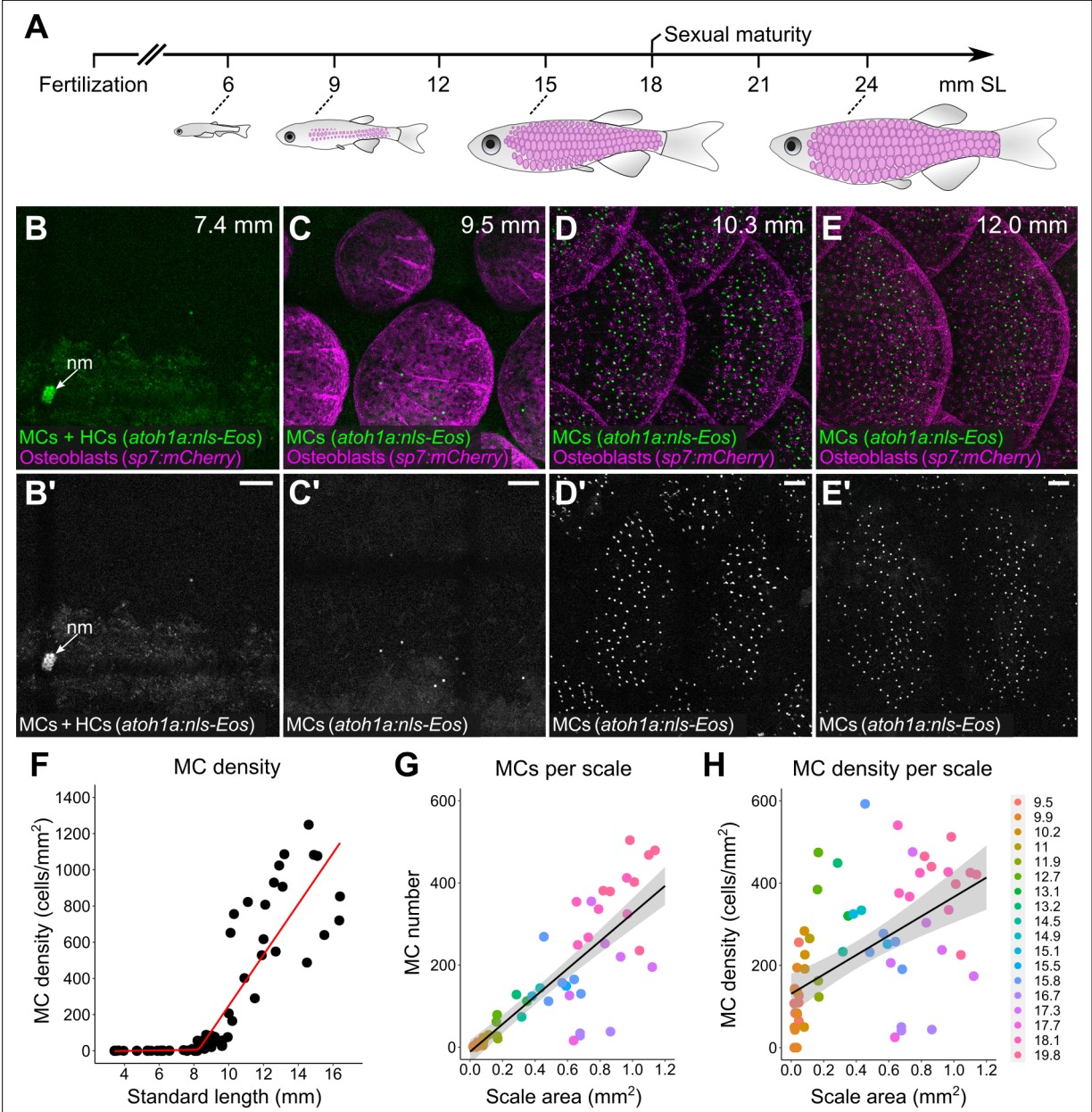

**Figure 7.** Merkel cells (MCs) develop concomitant with dermal appendage morphogenesis. (**A**) Abbreviated zebrafish developmental timeline relative to standard length (SL) in millimeters. Developing scales are drawn in magenta below the approximate corresponding stage. (**B–E**) Representative lateral confocal micrographs of MCs and osteoblasts along the trunk at the indicated stages. Note that MCs increase in number and density as scale-forming osteoblasts develop below the epidermis. nm, neuromast of the posterior lateral line. (**F**) Quantification of MC density according to SL. Each dot represents an individual fish. Data represent n=81 scales from N=52 fish. Line indicates segmented linear regression (breakpoint = 8.27 mm SL). (**G and H**) Quantification of the number (**G**) or density (**H**) of MCs relative to scale area. Each dot represents an individual scale. Data represent n=62 scales from N=18 fish. Dot colors represent animal SL as indicated in the legend. Shading indicates a 95% CI around the linear regression lines in **G** and **H**. Correlation coefficients ($R^2$): 0.08 (**F**, slope 1), 0.68 (**F**, slope 2), 0.73 (**G**), and 0.31 (**H**). F-statistics: 3.5 (**F**, slope 1), 83.9 (**F**, slope 2), 164.6 (**G**), and 28.31 (**H**). p-values: 0.07 (**F**, slope 1), <0.05 (**F**, slope 2), and <0.05 (**G and H**). Scale bars: 50 µm.

The online version of this article includes the following source data and figure supplement(s) for figure 7:

**Source data 1.** Datasheet for *Figure 7F*.

**Source data 2.** Datasheet for *Figure 7G and H*.

**Figure supplement 1.** MC number and density in relation to juvenile scale size.

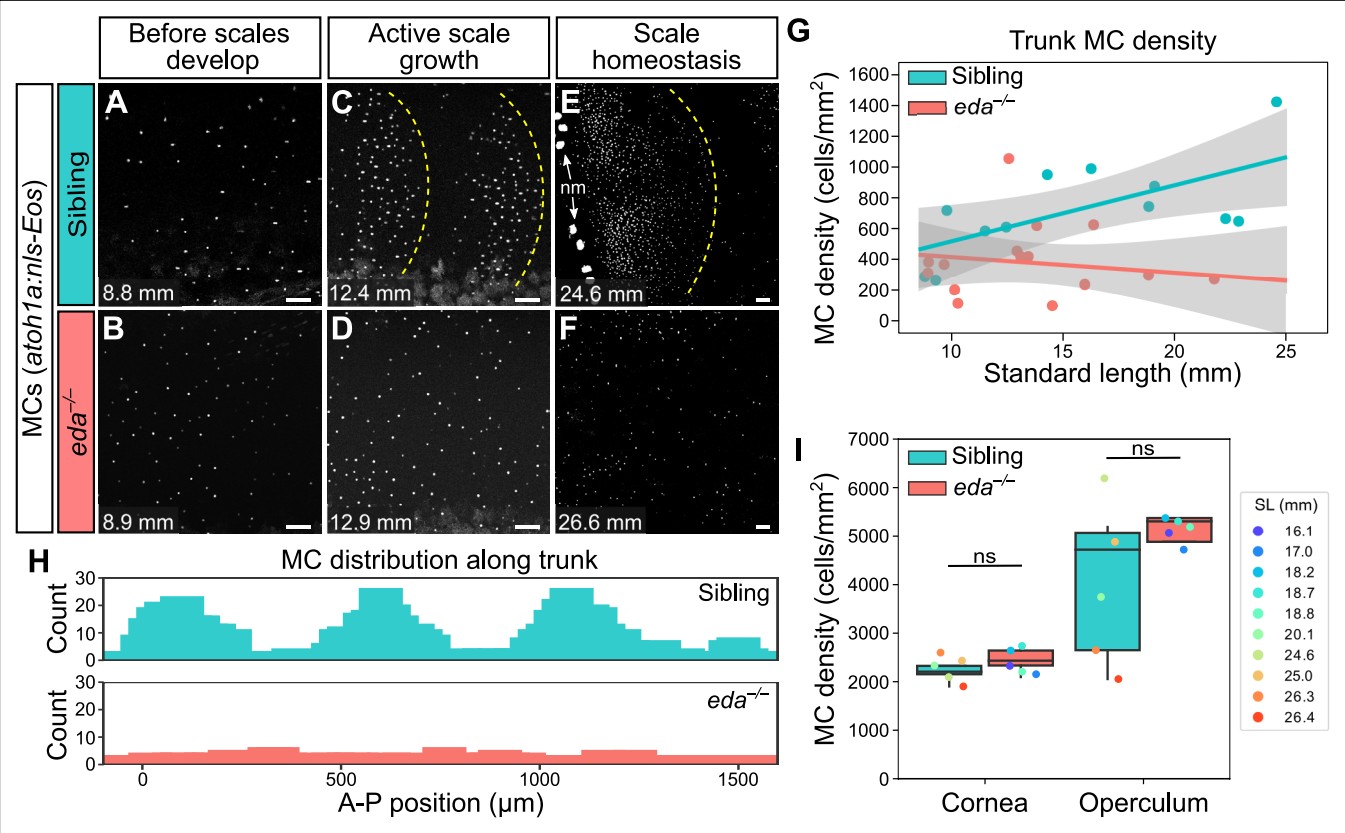

**Figure 8.** Loss of Eda signaling decreases Merkel cell (MC) density in trunk, but not facial skin. (**A–F**) Representative confocal images of MCs in the trunk of animals of the indicated genotypes at the indicated stages. Dotted yellow lines indicate posterior scale boundaries. nm, neuromasts of the posterior lateral line. (**G**) Quantification of MC density in the trunk skin relative to standard length (SL). Gray shading indicates a 95% CI around the linear regression lines. The difference between genotypes was significant above 12.5 mm SL ($p<0.05$, Johnson-Neyman Technique). Each dot represents an individual fish (N=16–18 fish/genotype). (**H**) Histograms of the distribution of trunk MCs along a rectangular segment encompassing three scales in a sibling and an identically sized region in an *eda* mutant (18–19 mm SL). (**I**) Boxplots of MC densities in the epidermis above the cornea or operculum in animals of the indicated genotypes. ns, not significant (cornea, $p=0.21$; operculum, $p=0.14$; Mann-Whitney test). Scale bars: 50 µm (**A–F**).

The online version of this article includes the following source data and figure supplement(s) for figure 8:

**Source data 1.** Datasheet for *Figure 8G*.

**Source data 2.** Datasheet for *Figure 8I*.

**Figure supplement 1—source data 1.** Datasheet for *Figure 8—figure supplement 1D and E*.

**Figure supplement 1.** *eda* mutants exhibit significantly decreased MC addition.

## Ectodysplasin signaling promotes trunk MC development

Since the appearance of MCs in the trunk epidermis tightly correlated with scale growth, we examined the consequences of blocking signals required for dermal appendage morphogenesis on MC development. Ectodysplasin (Eda) signaling regulates the formation of many types of skin appendages, including mammalian hair follicles and zebrafish scales (*Biggs and Mikkola, 2014*; *Harris et al., 2008*). To determine whether MC development required Eda-dependent signals, we measured MC density in animals homozygous for a presumptive null allele of *eda* that does not develop scales (*eda^dt1261/dt1261*; hereafter *eda^−/−*; *Harris et al., 2008*). Immediately prior to squamation, we found that there was no difference in MC density between *eda* mutants and sibling controls (*Figure 8A, B and G*). However, after the onset of squamation, *eda* mutants had significantly fewer MCs, a difference that persisted into adulthood (*Figure 8C–G*). In addition to the decrease in cell density, we observed a dramatic change in the spatial distribution of MCs across the epidermis in *eda* mutants compared to controls (*Figure 8H*). Specifically, in siblings, MCs appeared in patches corresponding to the location of the underlying scales (*Figure 8C and H*). By contrast, the MCs that developed in *eda* mutants were distributed relatively uniformly across the trunk (*Figure 8D and H*). Although we found a decrease in

MC density in the trunk skin of the mutants, we observed no change in MC density in the corneal or opercular epidermis (*Figure 8I*), suggesting that the reduced density was specific to the trunk skin. *eda* mutants lack fins at the stages analyzed (*Harris et al., 2008*), precluding analysis of these regions in the homozygous mutants.

The decreased MC density in *eda* mutant trunk skin could be due to decreased cell addition, increased cell loss, or a combination of the two. Using in vivo photoconversion, we found that the rate of MC addition was significantly reduced in *eda* mutants compared to siblings (*Figure 8—figure supplement 1A–D*). Additionally, the rate of cell loss was higher in mutants compared to siblings, although this change was not statistically significant (*Figure 8—figure supplement 1E*). Thus, our observations indicate that the decrease in MC cell density in *eda* mutants is mainly due to reduced MC production. Together, these data suggest that Eda signaling, either directly or indirectly, is required for MC development, maintenance, and distribution along the trunk.

## MC patterning is not predetermined along the trunk

Since blocking dermal appendage formation through loss of Eda signaling inhibited MC development, we next examined the consequences of altering dermal appendage size and shape on MC

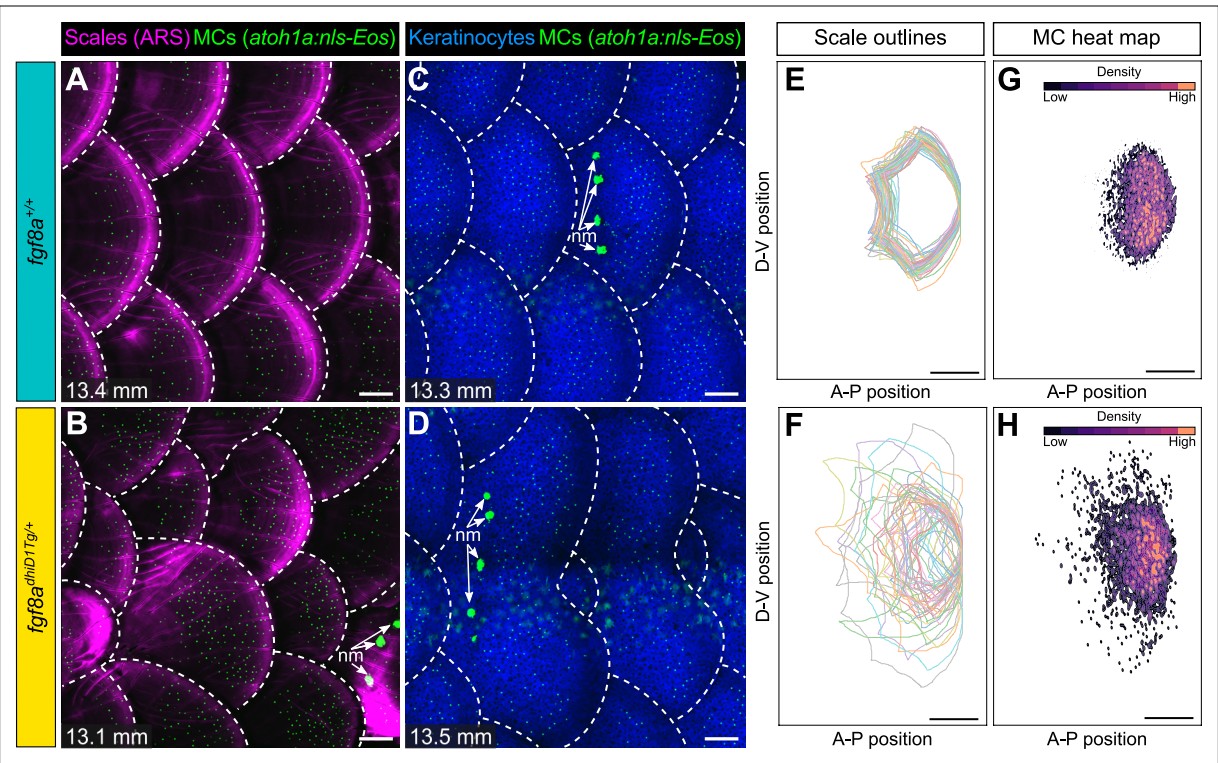

**Figure 9.** Merkel cell (MC) patterning is not predetermined along the trunk. (**A–D**) Representative images of juvenile animals of the indicated genotypes expressing an MC reporter and stained with Alizarin Red S (ARS) to visualize scales (**A and B**) or co-expressing MC and keratinocyte (*Tg(krt4:DsRed)*) reporters (**C and D**). Dotted lines indicate scale boundaries. nm, neuromasts of the posterior lateral line. (**E–H**) Tracings of scale outlines (**E and F**) and density plots of MC position (**G and H**) from juvenile animals (n=43–49 scales/genotype from N=10–13 fish/genotype; 11.6–14.7 mm standard length [SL]) of the indicated genotypes. Scale tracings were aligned at the dorsal-ventral midpoint of the posterior scale margin. Note the variability in scale shape and size and corresponding increased spread of MC position in *fgf8a*^dhiD1Tg/+ juveniles compared to sibling controls. Scale bars: 100 µm (**A–D**) and 200 µm (**E–H**).

The online version of this article includes the following source data and figure supplement(s) for figure 9:

**Source data 1.** Datasheet for *Figure 9G, H*, *Figure 9—figure supplement 1E*.

**Figure supplement 1.** *fgf8a*^dhiD1Tg/+ juveniles show altered dermal appendage size and shape, but not MC density.

**Figure supplement 1—source data 1.** Datasheet for *Figure 9—figure supplement 1A–C*.

**Figure supplement 1—source data 2.** Datasheet for *Figure 9—figure supplement 1D*.

**Figure supplement 1—source data 3.** Datasheet for *Figure 9—figure supplement 1E*.

patterning. Zebrafish scale morphogenesis is regulated by Fibroblast growth factor (FGF) signaling (*Aman et al., 2018*; *Daane et al., 2016*; *De Simone et al., 2021*; *Rohner et al., 2009*). To determine whether alterations to scale patterning impacted MC development, we examined animals heterozygous for an allele of *hagoromo* (*hag*; *fgf8a^{dhiD1Tg/+}*), which results in *fgf8a* overexpression in the post-embryonic skin due to a viral insertion near the *fgf8a* locus (*Amsterdam et al., 2009*). An independent allele of *hag* (*fgf8a^{dhi4000Tg/+}*) was previously shown to result in large, disorganized sheets of scale-forming osteoblasts during squamation (*Aman et al., 2018*). *fgf8a^{dhiD1Tg/+}* juveniles showed dramatic variability in scale size and shape, with both smaller and larger scales compared to the relatively uniformly patterned scales observed in sibling controls (*Figure 9A–D*; *Figure 9—figure supplement 1A–C*). We found no significant differences in MC density between the genotypes (*Figure 9—figure supplement 1D,E*). Nevertheless, the distribution of MCs tracked with the altered scale size and shape in the mutants (*Figure 9E–H*), suggesting the MC pattern is not predetermined within the trunk skin compartment.

## Discussion

Here, we discover a zebrafish epidermal cell type that we classify as an MC based on ultrastructural criteria (*Whitear, 1989*). We further present several lines of evidence that suggest zebrafish MCs share molecular, cellular, and lineage properties with mammalian MCs. First, we show that zebrafish MCs express the transcription factors Atoh1a and Sox2, the orthologs of which uniquely mark MCs in mammalian skin (*Maricich et al., 2009*; *Bardot et al., 2013*; *Perdigoto et al., 2014*; *Ostrowski et al., 2015*; *Van Keymeulen et al., 2009*). Second, zebrafish MCs extend numerous short, actin-rich microvilli and complex with somatosensory axons, classic morphological hallmarks of MCs (*Mihara et al., 1979*; *Smith, 1977*; *Toyoshima et al., 1998*). Our morphological observations support the interpretation that these cells are MCs rather than Merkel-like cells, which lack axon association and microvillar processes (reviewed by *Halata et al., 2003*). Third, Cre-based lineage tracing revealed that basal keratinocytes give rise to zebrafish MCs, akin to studies in mouse (*Morrison et al., 2009*; *Van Keymeulen et al., 2009*). Fourth, we demonstrate that zebrafish MCs contain neurosecretory machinery and express the neurotransmitter serotonin, the release of which has been proposed to regulate somatosensory responses to touch (*Chang et al., 2016*; *Chang and Gu, 2020*; *English et al., 1992*). Finally, we show that zebrafish MCs express the cation channel Piezo2, which is cell-autonomously required for MC mechanosensory function (*Ikeda et al., 2014*; *Maksimovic et al., 2014*; *Woo et al., 2014*). Importantly, our results extend previous histological studies of MCs in various teleost fish (*Lane and Whitear, 1977*; *Whitear, 1989*; *Zachar and Jonz, 2012*) by identifying the first genetically encoded reagents for the study of this cell type in zebrafish.

While our characterization revealed substantial similarities between mammalian and zebrafish MCs, we did observe anatomical differences in line with previous ultrastructural characterizations of teleost MCs (*Lane and Whitear, 1977*; *Whitear, 1989*). For example, the nuclei of mammalian MC are commonly lobulated (*Boulais et al., 2009*; *Cheng Chew and Leung, 1994*; *Moll et al., 2005*; *Tachibana and Nawa, 2002*). While we did not observe lobulation of zebrafish MC nuclei by TEM, we cannot rule out that serial sectioning or high-resolution reconstruction of nuclear shape would reveal lobulation. Mammalian MCs typically localize adjacent to basal keratinocytes (*Boot et al., 1992*; *Cheng Chew and Leung, 1994*; *Fradette et al., 1995*; *Mihara et al., 1979*; *Moll et al., 1996*; *Smith, 1977*), whereas zebrafish MCs appear in upper strata, typically beneath the periderm (*Figure 1D and G"*). As the majority of the analyses completed here focused on MCs found in the trunk epidermis, it will be intriguing to determine whether all MCs in different skin compartments in the juvenile and adult zebrafish share similar properties.

### Teleost MCs and somatosensory physiology

In mammalian skin, the MC-neurite complex regulates slowly adapting type I responses to light touch (*Iggo and Muir, 1969*; *Ikeda et al., 2014*; *Maksimovic et al., 2014*; *Maricich et al., 2009*; *Woo et al., 2014*). Although physiological studies of somatosensory responses in adult zebrafish have not been reported, extracellular recordings in adult rainbow trout demonstrate that a subset of somatosensory neurons exhibit slowly adapting responses to mechanical skin stimulation (*Ashley et al., 2007*; *Ashley et al., 2006*; *Sneddon, 2003*). We postulate that the slowly adapting responses to mechanical

skin stimulation in adults require MCs. Nevertheless, the exact physiological roles of teleost MCs in regulating somatosensory responses and resulting behaviors remain unknown and will require the development of tools to selectively ablate and activate MCs. Interestingly, recordings from zebrafish Rohon-Beard neurons, a transient larval somatosensory population, suggest they have rapidly, but not slowly, adapting mechanosensory responses (*Katz et al., 2021*). Together these studies correlate with our finding that MCs develop at post-larval stages and suggest that the teleost somatosensory system undergoes significant functional maturation during the juvenile period.

What are the subtypes of somatosensory neurons in fish, and how do they correspond to MC innervation? Several studies have identified molecularly distinct subsets of somatosensory neurons through mRNA, protein, and transgene expression analysis in zebrafish larvae (*Faucherre et al., 2013*; *Gau et al., 2017*; *Gau et al., 2013*; *Kucenas et al., 2006*; *Palanca et al., 2013*; *Pan et al., 2012*; *Patten et al., 2007*; *Slatter et al., 2005*). Adult trout somatosensory neurons have been classified based on their responses to mechanical, chemical, and thermal stimuli (*Ashley et al., 2007*; *Ashley et al., 2006*; *Sneddon, 2003*). However, to date, a detailed molecular characterization of the diversity of somatosensory subtypes present in adult fish has not been performed. Our data suggest that somatosensory neurons expressing reporters for *p2rx3a*, *p2rx3*b, or *trpa1b* innervate MCs. Whether these neurons represent a dedicated class of MC-innervating neurons remains unknown. The development of Cre drivers for specific somatosensory subtypes (*Bai et al., 2015*; *Li et al., 2011*; *Luo et al., 2009*; *Rutlin et al., 2014*; *Zylka et al., 2005*) and single-cell transcriptional profiling (*Sharma et al., 2020*; *Usoskin et al., 2015*; *Zeisel et al., 2018*) has been fruitful in characterizing the diversity of somatosensory neurons in mammals. The application of these technologies to the teleost somatosensory system is an interesting avenue for further investigation.

## MC lineage and homeostasis

The developmental lineage of MCs has been a long-standing question with both epidermal and neural crest origins posited (*Hartschuh et al., 1986*). Our Cre-based lineage tracing identified basal keratinocytes as MC progenitors. These results extend previous studies in the zebrafish epidermis showing that basal keratinocytes serve as precursors for diverse post-larval cell types, including periderm and immune cells (*Lee et al., 2014*; *Lin et al., 2019*). Although previous work in mouse unambiguously identified *keratin 14*-expressing basal keratinocytes as MC precursors (*Morrison et al., 2009*; *Van Keymeulen et al., 2009*), the precise nature of murine MC progenitors varies across skin compartments (*Nguyen et al., 2019*). Future studies characterizing the molecular properties and cellular behaviors of zebrafish MC precursors will be informative for identifying conserved properties of skin stem cells.

The turnover of MCs in mammalian skin has been a source of controversy. Several studies reported that MC numbers fluctuate with the natural hair cycle in mouse (*Marshall et al., 2016*; *Moll et al., 1996*; *Nakafusa et al., 2006*). By contrast, *Wright et al., 2017* found no evidence for changes in MC density based on stages of the hair cycle and demonstrated that MCs could live for months. These types of analyses in murine skin have relied either on histology, which limits tissue sampling, or required the use of advanced (2-photon) microscopy in combination with hair shaving, a mild form of skin injury. Using photoconversion and confocal imaging, we non-invasively tracked individual MCs during normal skin homeostasis in vivo for weeks. We found that trunk MCs have a steady turnover in adult animals, with a half-life of approximately 1 month. Additionally, this further distinguishes MCs from hair cells in the adult lateral line, which have a shorter half-life (*Cruz et al., 2015*). Whether MC turnover varies at different stages of development, across skin compartments, or following skin insults will require further study.

## MC distribution and patterning

Regionally specific sensory structures allow our skin to distinguish tactile inputs with remarkable acuity (*Corniani and Saal, 2020*). For example, MC densities vary greatly across human skin compartments, with the highest numbers found in particularly sensitive regions such as fingertips and lips (*Boot et al., 1992*; *Lacour et al., 1991*). We observed that MCs populate several major skin compartments and have regional-specific densities in adult zebrafish, with the highest densities found in the face (corneal and opercular epidermis). We speculate this may bestow the juvenile and adult skin with the ability

to detect innocuous tactile inputs across almost the entire body surface, with perhaps the greatest sensitivity along facial structures.

Although most studies of MC development have centered on the formation of MC aggregates in the touch domes of murine hairy skin, MCs are found in a range of distribution patterns in other types of skin. For example, MCs are found as dispersed, single cells arrayed across the skin of human toe pads (*Boot et al., 1992*). Similarly, we found that MCs have a dispersed, rather than clustered, pattern in all skin compartments examined. Few studies have addressed how MCs adopt specific distributions, and zebrafish presents a promising model to understand mechanisms of MC pattern formation in vivo.

## Dermal appendages and MC development

Our developmental analysis showed that trunk MC density rapidly increases during dermal appendage morphogenesis. Previous genetic analysis in mouse hairy skin revealed that MC development requires Eda signaling (*Vielkind et al., 1995*; *Xiao et al., 2016*). We show that zebrafish *eda* mutants have decreased MC density in the trunk, but not the facial, skin. These observations suggest that MC development in mouse and zebrafish likely share similar genetic pathways, akin to the shared molecular and cellular mechanisms that regulate dermal appendage formation (*Aman et al., 2018*; *Biggs and Mikkola, 2014*; *Daane et al., 2016*; *Harris et al., 2008*; *Rohner et al., 2009*). They further support a model whereby MC development requires compartment-specific signals, akin to recent observations on MC development in mouse hairy and glabrous skin (*Nguyen et al., 2019*). By taking advantage of the ability to image large skin areas in intact zebrafish, we show that *eda* mutants have altered MC distribution compared to controls. Furthermore, we use in vivo photoconversion to demonstrate the reduction in MC density is largely due to decreased production but also reflects slightly increased cell loss. Further investigations are required to determine whether Eda signaling directly regulates the differentiation of MC progenitors. Alternatively, since *eda* mutants lack scales (*Harris et al., 2008*) and have decreased epidermal innervation (*Rasmussen et al., 2018*), MC development may require scale- and/or somatosensory neuron-derived signals. Finally, we note that trunk MCs are not completely absent in *eda* mutants, suggesting that a subset of MCs develop independent of Eda signaling.

We found that a gain-of-function allele of *fgf8a* leads to a change in the overall size and shape of scales. Intriguingly, the MC distribution modifies to accommodate the altered scale size and shape in the *fgf8a* mutants but retains the same MC density as sibling controls. This result suggests that the number of MCs per scale is not predetermined but rather is titrated relative to appendage size. How are MCs able to populate the much larger scales in *fgf8a* mutants? Does the size of the MC progenitor domain expand with increases in scale size? Are MCs, or their progenitors, able to migrate to their final destination? Distinguishing between these possibilities will require tracking the behaviors of MCs and their progenitors in vivo.

## Summary

Our results establish a promising new system to investigate MC biology. This model will allow for the identification of deeply conserved mechanisms used to regulate vertebrate MC biology. Furthermore, the advantages of zebrafish—such as non-invasive in vivo imaging, genetic and chemical screens, and high regenerative capacity—will complement the strengths of existing rodent models. Specifically, the ability to track individual cells over time has the potential to answer key and long-standing questions surrounding MC biology, including how the MC-neurite relationship is established, how MCs interact with neighboring cell types, and their progenitor dynamics. Addressing these questions, as well as potential novel insights provided by the zebrafish system, represent exciting directions for future research.

## Materials and methods

**Key resources table**

| Reagent type (species) or resource | Designation | Source or reference | Identifiers | Additional information |
|---|---|---|---|---|
| Antibody | Anti-Serotonin (rabbit polyclonal) | MilliporeSigma | Cat #: S5545, RRID:AB_477522 | (1:1000) |

*Continued on next page*

*Continued*

| Reagent type (species) or resource | Designation | Source or reference | Identifiers | Additional information |
|---|---|---|---|---|
| Antibody | Anti-Sv2 (mouse monoclonal) | DSHB; (*Buckley and Kelly, 1985*) | Cat #: SV2, RRID:AB_2315387 | (1:50) |
| Antibody | Anti-Sox2 (rabbit polyclonal) | GeneTex | Cat #: GTX124477, RRID:AB_11178063 | (1:500) |
| Antibody | Anti-Fluorescein Polyclonal Antibody, POD Conjugated (sheep polyclonal) | Roche | Cat #: 11426346910, RRID:AB_840257 | (1:2000) |
| Antibody | Anti-GFP (rabbit polyclonal) | Thermo Fisher Scientific | Cat #: A11122, RRID:AB_221569 | (1:1000) |
| Antibody | zn-12 (mouse monoclonal) | Zebrafish International Resource Center | Cat #: zn-12, RRID:AB_10013761 | (1:200) |
| Antibody | Anti-Rabbit Alexa Fluor 647 (goat polyclonal) | Thermo Fisher Scientific | Cat #: A32733, RRID:AB_2633282 | (1:500) |
| Antibody | Anti-Mouse Alexa Fluor 647 (goat polyclonal) | Thermo Fisher Scientific | Cat #: A32728, RRID:AB_2633277 | (1:500) |
| Antibody | Anti-Rabbit Alexa Fluor 568 (goat polyclonal) | Thermo Fisher Scientific | Cat #: A-11036, RRID:AB_10563566 | (1:500) |
| Antibody | Anti-Mouse Alexa Fluor 568 (goat polyclonal) | Thermo Fisher Scientific | Cat #: A-11031, RRID:AB_144696 | (1:500) |
| Antibody | Anti-Rabbit Alexa Fluor 488 (goat polyclonal) | Thermo Fisher Scientific | Cat #: A32731, RRID:AB_2633280 | (1:500) |
| Commercial assay and kit | TSA Plus Cyanine 5 | Akoya Biosciences | Cat #: NEL705A001KT | (1:50) |
| Sequence-based reagent | *piezo2* in situ probe (originally referred to as *piezo2b*) | *Faucherre et al., 2013* | N/A | |
| Sequence-based reagent | *atoh1a* gRNA, 5'-GGA GAC TGA ATA AAG TTA TG-3' | *Pickett et al., 2018* | N/A | |
| Sequence-based reagent | Mbait gRNA, 5'-GGC TGC TGC GGT TCC AGA GGT GG-3' | *Kimura et al., 2014* | N/A | |
| Sequence-based reagent | Zebrafish *piezo2* HCR v3.0 probe | Molecular Instruments | N/A | Used at 2 pmol |
| Chemical compound and drug | MS-222 | MilliporeSigma | Cat #: E10521 | |
| Chemical compound and drug | (Z)–4-Hydroxytamoxifen (4-OHT) | MilliporeSigma | Cat #: H7904 | Used at 10 µM |
| Strain and strain background (*Danio rerio*) | AB (Wild-Type) | Zebrafish International Resource Center | ZIRC Cat# ZL1, RRID:ZIRC_ZL1 | |
| Genetic reagent (*Danio rerio*) | *Tg(actb2:LOXP-BFP-LOXP-DsRed)* | *Kobayashi et al., 2014* | *Tg(actb2:LOXP-BFP-LOXP-DsRed)$^{sd27Tg}$*, ZFIN: ZDB-TGCONSTRUCT-141111–5 | |
| Genetic reagent (*Danio rerio*) | *Tg(atoh1a:nls-Eos)* | *Pickett et al., 2018* | *Tg(atoh1a:nls-Eos)$^{w214Tg}$*, ZFIN: ZDB-TGCONSTRCT-190701–2 | |
| Genetic reagent (*Danio rerio*) | *Tg(atoh1a:lifeact-EGFP)* | This study | *Tg(atoh1a:lifeact-EGFP)$^{w259Tg}$* | |
| Genetic reagent (*Danio rerio*) | *TgBAC(ΔNp63:Cre-ERT2)* | This study | *TgBAC(ΔNp63:Cre-ERT2)$^{w267Tg}$* | |
| Genetic reagent (*Danio rerio*) | *Tg(sox10:Cre)* | *Kague et al., 2012* | *Tg(Mmu.Sox10-Mmu.Fos:Cre)$^{zf384}$*, ZFIN: ZDB-TGCONSTRCT-130614–2 | |
| Genetic reagent (*Danio rerio*) | *Gt(ctnna-citrine)* | *Trinh et al., 2011* | *Gt(ctnna-citrine)$^{ct3aGt}$*, ZFIN: ZDB-ALT-111010–23 | |

*Continued on next page*

*Continued*

| Reagent type (species) or resource | Designation | Source or reference | Identifiers | Additional information |
|---|---|---|---|---|
| Genetic reagent (*Danio rerio*) | *Tg(sp7:mCherry)* | *Singh et al., 2012* | *Tg(Ola.Sp7:mCherry-Eco.NfsB)*<sup>pd46Tg</sup>, ZFIN: ZDB-TGCONSTRUCT-120503–4 | |
| Genetic reagent (*Danio rerio*) | *Tg(p2rx3a>mCherry)* | *Palanca et al., 2013* | *Tg(Tru.P2rx3a:LEXA-VP16,4xLEXOP-mCherry)*<sup>la207Tg</sup>, ZFIN: ZDB-TGCONSTRCT-130307–1 | |
| Genetic reagent (*Danio rerio*) | *Tg(trpa1b:EGFP)* | *Pan et al., 2012* | *TgBAC(trpa1b:EGFP)*<sup>a129Tg</sup>, ZFIN: ZDB-TGCONSTRCT-120208–2 | |
| Genetic reagent (*Danio rerio*) | *Tg(p2rx3b:EGFP)* | *Kucenas et al., 2006* | *Tg(p2rx3b:EGFP)*<sup>sl1Tg</sup>, ZFIN: ZDB-TGCONSTRCT-070117–110 | |
| Genetic reagent (*Danio rerio*) | *Tg(krt4:DsRed)* | *Rieger and Sagasti, 2011* | *Tg(krt4:DsRed)*<sup>la203Tg</sup>, ZFIN: ZDB-TGCONSTRCT-120127–5 | |
| Genetic reagent (*Danio rerio*) | *eda*<sup>dt1261</sup> | *Harris et al., 2008* | *eda*<sup>dt1261</sup>, ZFIN: ZDB-ALT-090324–1 | |
| Genetic reagent (*Danio rerio*) | *fgf8a*<sup>dhiD1Tg/+</sup> | *Amsterdam et al., 2009* | *fgf8a*<sup>dhiD1Tg/+</sup>, ZFIN: ZDB-ALT-010427–4 | |
| Software and algorithm | FIJI | http://fiji.sc | RRID:SCR_002285 | |
| Software and algorithm | Imaris | Bitplane | RRID:SCR_007370 | |
| Other | Fetal bovine serum | Gibco | Cat #: 10082–139 | |
| Other | Normal goat serum | Abcam | Cat #: ab7481, RRID:AB_2716553 | |
| Other | DAPI | MilliporeSigma | Cat #: 508741 | Used at 5 ng/µl |
| Other | AM1-43 | Biotinium | Cat #: 70024 | Used at 15 µm |
| Other | Alizarin Red S | ACROS Organics | Cat #: 400480250 | Used at 0.01% |
| Other | Proteinase K | Thermo Fisher Scientific | Cat #: 100005393 | Used at 0.1 mg/ml |
| Other | Hoechst 3342 | Thermo Fisher Scientific | Cat #: H3570 | Used at 5 ng/µl |

## Materials availability

Strains and DNA constructs generated in this study are available from the corresponding author upon request.

## Animals

### Zebrafish

Zebrafish were housed at 26–27°C on a 14/10 hr light cycle. See Key resources table for strains used in this study. Animals of either sex were used. Zebrafish were staged according to SL (*Parichy et al., 2009*). SL of fish was measured using the IC Measure software (The Imaging Source) on images captured on a Stemi 508 stereoscope (Zeiss) equipped with a DFK 33UX264 camera (The Imaging Source). All zebrafish experiments were approved by the Institutional Animal Care and Use Committee at the University of Washington (Protocol: #4439–01).

### Creation of *Tg(atoh1a:Lifeact-EGFP)*

*Tg(atoh1a:Lifeact-EGFP)*<sup>w259Tg</sup> was generated by CRISPR-mediated knock-in as previously described (*Kimura et al., 2014*). A donor plasmid containing the Mbait, minimal *hsp70l* promoter, *Lifeact-EGFP*, and *bgh poly(A)* sequences was created using Gibson assembly. The insertion was targeted 372 bp upstream of the endogenous *atoh1a* coding sequence using a previously published guide RNA (gRNA; *Pickett et al., 2018*). The *Mbait-hsp70l-Lifeact-EGFP* plasmid, Mbait and *atoh1a* gRNAs, and Cas9 protein were prepared and injected into single-cell embryos of the AB strain as previously described (*Thomas and Raible, 2019*). Larvae were screened for Lifeact-EGFP expression at 3 dpf and raised to adulthood. A founder adult was identified and outcrossed to generate a stable transgenic line.

## Creation of *TgBAC(ΔNp63:Cre-ERT2)* and induction with 4-OHT

The *ΔNp63:Cre-ERT2* bacterial artificial chromosome (BAC) was created by modifying the previously generated BAC DKEY-263P13-iTol2-amp (*Rasmussen et al., 2015*). The predicted *ΔNp63* start codon was replaced by a *Cre-ERT2-pA-KanR* cassette that contained a zebrafish codon-optimized *Cre-ERT2* (*Kesavan et al., 2018*) using a previously described protocol (*Suster et al., 2011*). *TgBAC(ΔNp63:Cre-ERT2)*[w267Tg] was created by injecting *tol2* mRNA, which was transcribed from pCS2-zT2TP (*Suster et al., 2011*), and BAC DNA into one-cell stage embryos and screening adults for germline transmission. To activate Cre-ERT2, 1 dpf embryos were treated with 10 μM 4-OHT for 24 hr. 4-OHT was prepared as described (*Felker et al., 2016*).

## Mutant identification and analysis

*eda* mutants and siblings were sorted by visible phenotype starting at 7 mm SL. Mutants were grown separately from siblings. *fgf8a*[dhiD1Tg/+] fish were identified based on altered scale patterning and/or pigmentation (*Kawakami et al., 2000*).

## Imaging and photoconversion

### Electron microscopy

Isolated scales were prepared for TEM as described (*Sire et al., 1997b*), with the following modifications: after dehydration, scales were treated with propylene oxide (PO), infiltrated with PO:Eponate 12, and embedded in Eponate 12. Semithin sections (0.2 μm) stained with toluidine blue were used for orientation. Thin sections (50 nm) were placed on Formvar coated copper slot grids, stained with saturated uranyl acetate and Reynolds' lead citrate, and examined on a JEOL 100 CX at 60 kV or a Philips CM100 at 80 kV.

### Confocal image acquisition

Confocal z-stacks were collected using a A1R MP+ confocal scanhead mounted on an Ni-E upright microscope (Nikon) using a 16× water dipping objective (NA 0.8) for live imaging or 40× oil immersion objective (NA 1.3) for fixed image acquisition. Images acquired in resonant scanning mode were post-processed using the denoise.ai function in NIS-Elements (Nikon). For live imaging, zebrafish were anesthetized in a solution of 0.006–0.012%buffered MS-222 in system water for 5 min. Anesthetized fish were mounted in a custom imaging chamber, partially embedded in 1% agarose, and covered with MS-222-containing solution. For *Video 1*, a FLUOVIEW FV3000 scanning confocal microscope (Olympus) equipped with a 100× objective (NA 1.49) was used to collect a z-stack which was 3D rendered with Imaris (Bitplane).

### Whole animal photoconversion

Prior to imaging, *Tg(atoh1a:nls-Eos)* zebrafish were exposed to light from a UV LED flashlight (McDoer) for 15 min in a reflective chamber constructed from a styrofoam box lined with aluminum foil. A similar lateral region of the trunk was imaged over subsequent days identified by approximate body position below the dorsal fin and relative to underlying pigment stripes.

### Regional photoconversion

After anesthetization and mounting as described above, the *Tg(atoh1a:nls-Eos)* reporter was photoconverted using the stimulation program of NIS-Elements with the 405 nm laser at 14–18% power for 30–45 s within a 500×500 pixel region of interest with an area of 67,055 μm$^2$. The same lateral region of the trunk was imaged over subsequent days identified by body position under the dorsal fin, position relative to underlying pigment stripes, and presence of photoconverted cells. Animals that died over the course of the experiment were excluded from further analysis.

## Staining

### Alizarin Red S staining

To visualize mineralized bone, live animals were stained for 15 min in a solution of 0.01% (wt/vol) Alizarin Red S dissolved in system water and subsequently rinsed 3×5 min in system water prior to imaging as described (*Bensimon-Brito et al., 2016*).

## Antibody staining

Zebrafish were anesthetized in a solution of 0.012% MS-222 in system water for 5 min. Using metal forceps, up to 10 scales were removed from the lateral side of the trunk in the region below the dorsal fin. Scales were fixed in 4% paraformaldehyde (PFA)/PBS at 4°C overnight. Scales were washed 4×5 min in 1× PBS +0.3% triton-X (PBST) at room temperature and then blocked for 1.5 hr with PBST containing 5% normal goat serum. Incubation with primary antibodies occurred at 4°C overnight, followed by 4×15 min washes in PBST. Scales were incubated in appropriate secondary antibodies for 2 hr at room temperature and washed 4×15 min in PBST. To label nuclei, scales were incubated with DAPI for 5 min at 4°C and washed in PBST 4×5 min at room temperature. Scales were mounted between a microscope slide and coverslip in Prolong gold. All steps were performed on a rotating platform.

## AM1-43 staining

Scales were removed from adult *Tg(atoh1a:nls-Eos)* zebrafish as described above and placed into the center of a petri dish. 1 ml of L-15 media was added to the dish containing newly plucked scales no longer than 2 min after the scales had been removed. 1.5 µl of 10 mM AM1-43 was added to the dish for a final concentration of 15 µM AM1-43. Scales were incubated for 5 min in this solution to allow for incorporation. Prior to confocal imaging, regional photoconversion of nls-Eos was carried out as described above.

## Fluorescent in situ hybridization

*piezo2* antisense RNA was transcribed in vitro from a previously generated plasmid (*Faucherre et al., 2013*) using SP6 and fluorescein-dUTP. The FISH protocol for adult zebrafish scales was previously described (*Lin et al., 2019*). Briefly, scales from *Tg(atoh1a:Lifeact-EGFP)* adults were plucked and fixed in 4% PFA overnight at 4°C then washed three times with 1× PBS +0.1% Tween-20 (PBSTw). Scales were dehydrated in sequential washes of 75% PBSTw:25% methanol (MeOH), 50% PBSTw:50% MeOH, 25%PBSTw:75%MeOH, then placed in 100% MeOH at –20°C overnight. Scales were rehydrated in sequential washes of 25% PBSTw:75% MeOH, 50% PBSTw:50% MeOH, 75%PBSTw:25% MeOH, then washed 3× in PBSTw. Scales were treated with 0.1 mg/ml proteinase K for 5 min, then re-fixed in 4% PFA for 20 min. Scales were washed once in PBSTw, washed once in 50% PBSTw:50% hybridization buffer, then incubated in hybridization buffer for 2 hr at 65°C. Scales were incubated in hybridization buffer with probe (~1 ng/µl) overnight at 65°C. Scales were sequentially washed at 65°C in 75% hybridization buffer:25% 2× SSC +0.1% Tween20 (SSCT), 50% hybridization buffer:50% 2× SSCT, 25% hybridization buffer:75% 2xSSCT, followed by three washes at room temperature in 2× SSCT, followed by three washes in 0.2× SSCT. Scales were then washed 3× in 1× PBS +0.2% Triton X-100 (PBSTr), then blocked for 2 hr in PBSTr +5% fetal bovine serum. Scales were incubated in blocking buffer with an anti-fluorescein peroxidase (POD) conjugated antibody (1:2000) overnight at 4°C. Scales were washed 6× in PBSTr, followed by staining with Tyramide Signal Amplification (TSA) Plus Cyanine 5 (1:50 dilution) for 10 min.

Following FISH, scales were incubated in PBSTr +10% NGS for 2 hr at room temperature. Scales were stained with an anti-GFP antibody (1:1000) in PBSTr +10% NGS overnight at 4°C. Scales were washed in PBSTr, then incubated in secondary antibodies (1:200) for 2 hr at room temperature. Scales were washed in PBSTr, stained with Hoechst (3.24 nM) for 10 min at room temperature, washed in PBSTr, mounted under coverslips in ProLong Gold, and imaged.

## Hybridization chain reaction

A custom *piezo2* probe set (set size: 20; amplifier: B3) was ordered using accession number XM_021468270.1. For HCR on adult zebrafish scales, minor alterations were made to a previously described protocol (*Ibarra-García-Padilla et al., 2021*). Briefly, scales from *Tg(atoh1a:Lifeact-EGFP)* adults were plucked and fixed in 4% PFA overnight at 4°C, with 20 scales per 1.5 ml tube. Scales were washed 3× in 1× PBS and then dehydrated and permeabilized with 2×10 min washes in 100% methanol. The samples were stored at –20°C overnight. To rehydrate the samples, a series of graded MeOH/PBSTw washes were used for 5 min each: 75% MeOH:25% 1× PBSTw, 50% MeOH:50% 1× PBSTw, 25% MeOH:75% 1× PBSTw, and finally 2× washes in 100% 1× PBSTw. To further permeabilize

the scales, samples were incubated in 10 ug/ml Proteinase K diluted in 1× PBSTw for 10 min. Samples were quickly washed 3× in 1× PBSTw, and then post-fixed with 4% PFA for 20 min. After post-fixation, samples underwent 5×5 min washes with 1× PBSTw. Samples were then pre-hybridized with Molecular Instruments HCR hybridization buffer at 37°C for 30 min. After pre-hybridization, samples were incubated with 2 pmol of the probe set diluted in hybridization buffer for 16 hr at 37°C. To remove the probe mixture solution, samples were washed 4× for 15 min each with probe wash buffer at 37°C. Samples were washed 2× for 5 min with 5× SSC +0.1% Tween-20 and then treated with probe amplification buffer for 30 min at room temperature. Samples were washed into hairpin amplification buffer containing snap cooled amplifier hairpins and were incubated at room temperature, protected from light, overnight. Samples were then washed with successive 5× SSC +0.1% Tween-20 washes: 2× washes for 5 min, 2× washes for 30 min, and 1× wash for 5 min. Finally, samples underwent 3×5 min washes with 1× PBSTw. Anti-GFP staining was performed as described above.

## Image analysis

### Axon contact quantification

Innervation of *Tg(atoh1a:nls-Eos)*-expressing cells was scored using a custom ImageJ macro. The macro identified the centroid of each *atoh1a+* cell using the '3D project' function and then created a 3D sphere that was 10% larger than the maximum nuclear diameter. A cell was scored as innervated if an axon passed within the sphere. In some cases, the *Tg(atoh1a:nls-Eos)* reporter was photoconverted prior to image acquisition as described above.

### Cell density analysis

Maximum intensity projections of confocal z-stacks were converted to 8-bit images and thresholded in ImageJ. Cell density was quantified using the 'Analyze particles' function of ImageJ. For low-magnification quantification of MC cell density across the trunk of *fgf8a^{dhiD1Tg/+}* and siblings, tiled images were collected that included multiple scales per region. Cell density was quantified as described above using ImageJ. For high-magnification cell density quantification in the epidermis directly above scales, a small region centered in the epidermis of each full scale in view and positioned based on scale lobe was quantified. For scales with multiple lobes, a density measurement was collected from the center of each lobe and averaged.

### Recombination efficiency analysis

Maximum intensity projections of the DsRed channel of confocal z-stacks for the basal keratinocyte Cre lineage tracing experiments were converted to 8-bit images and thresholded using the 'Huang' preset in ImageJ. The '%Area' of the thresholded image was determined using the 'Measure' function and taken to represent recombination efficiency.

### Statistical analysis

Statistical tests used are listed in individual figure legends. Plots were created using R or Python. N refers to either the number of individual fish or the number of individual scales where appropriate and is specified in figure legends. For boxplots, each graph possesses a black line within the box that represents the median, two hinges for the first and third quartiles, two whiskers that extend no further than 1.5× (interquartile range) from the adjacent hinge, and outlying points plotted individually beyond the whiskers. Segmented linear regression in *Figure 7F* was performed using the R function 'segmented' with parameter npsi = 1.

## Acknowledgements

We thank the LSB Aquatics staff for animal care; Wai Pang Chan and Marianne Cilluffo for TEM support; the labs of Ajay Dhaka, Jacqueline Lees, and Alvaro Sagasti for sharing zebrafish stocks; the lab of Chris Joplin for sharing the *piezo2* plasmid. The authors are grateful to all members of the Rasmussen lab for discussion, technical assistance, and support. This work was funded in part by a Postdoctoral Fellowship (#2011008) from the National Science Foundation to TLB, a Graduate Research Fellowship (DGE-2140004) from the National Science Foundation to EWC, R01HD107108 from the Eunice Kennedy Shriver National Institute of Child Health and Human Development to JPR, A153025 from

the University of Washington Research Royalty Fund to JPR, and a New Investigator Award from the University of Washington/Fred Hutchinson Cancer Research Center Cancer Consortium, which is supported by the NIH/NCI Cancer Center Support Grant P30 CA015704, to JPR. JPR is a Washington Research Foundation Distinguished Investigator.

## Additional information

### Funding

| Funder | Grant reference number | Author |
| --- | --- | --- |
| National Science Foundation | 2011008 | Tanya L Brown |
| National Science Foundation | DGE-2140004 | Evan W Craig |
| Eunice Kennedy Shriver National Institute of Child Health and Human Development | R01HD107108 | Jeffrey P Rasmussen |
| University of Washington Research Royalty Fund | A153025 | Jeffrey P Rasmussen |
| Cancer Consortium | NIH/NCI Cancer Center Support Grant P30 CA015704 | Jeffrey P Rasmussen |

The funders had no role in study design, data collection and interpretation, or the decision to submit the work for publication.

### Author contributions

Tanya L Brown, Conceptualization, Formal analysis, Funding acquisition, Investigation, Visualization, Methodology, Writing – original draft, Writing – review and editing; Emma C Horton, Software, Formal analysis, Investigation, Visualization, Methodology, Writing – review and editing; Evan W Craig, Formal analysis, Funding acquisition, Investigation, Visualization, Methodology; Camille EA Goo, Erik C Black, Formal analysis, Investigation, Visualization, Methodology, Writing – review and editing; Madeleine N Hewitt, Resources; Nathaniel G Yee, Everett T Fan, Formal analysis, Investigation, Visualization; David W Raible, Supervision; Jeffrey P Rasmussen, Conceptualization, Formal analysis, Supervision, Funding acquisition, Validation, Investigation, Visualization, Writing – original draft, Project administration, Writing – review and editing

### Author ORCIDs

Tanya L Brown  http://orcid.org/0000-0001-9554-178X
Emma C Horton  http://orcid.org/0000-0001-9730-7380
Camille EA Goo  http://orcid.org/0000-0002-9118-4006
Erik C Black  http://orcid.org/0000-0002-2333-8923
Madeleine N Hewitt  http://orcid.org/0000-0002-4387-327X
David W Raible  http://orcid.org/0000-0002-5342-5841
Jeffrey P Rasmussen  http://orcid.org/0000-0001-6997-3773

### Ethics

All zebrafish experiments were approved by the Institutional Animal Care and Use Committee at the University of Washington (Protocol: #4439-01).

### Decision letter and Author response

Decision letter https://doi.org/10.7554/eLife.85800.sa1
Author response https://doi.org/10.7554/eLife.85800.sa2

# Additional files

## Supplementary files
• MDAR checklist

## Data availability
All data generated or analyses during this study are included in the manuscript and supporting files. Source data files have been provided as indicated.

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
