## [Editor Report]

The authors describe and characterize the touch system in zebrafish as a new model to study Merkel cell development and maintenance. The study demonstrates that the zebrafish touch system shares many characteristics with its mammalian counterpart, including developmental origin, innervation, and molecular characteristics while allowing in vivo analysis of specification, development, and maintenance. This study is the foundation for future detailed cellular and molecular analyses of the touch sensory system and will be of interest to developmental biologists and neuroscientists studying stem cells, regeneration, and aging.

---

## [Decision Letter]

[Editors' note: this paper was reviewed by Review Commons.]

---

## [Author Response]

We thank all the reviewers for their positive and constructive comments on the manuscript. We address each of the reviewers’ comments point-by-point below. The major revisions include:

Improved statistical details and attention to subjective language throughout.New TEM data included in the new Figure 1—figure supplement 1 to illustrate the drastic ultrastructural differences between MCs and neighboring epidermal cells.Inclusion of an estimate of the “recombination efficiency” of our keratinocyte lineage trace in Figure 4.Additional quantification of MC density in the different body regions (Figure 6) and prior to squamation (Figure 7F). Imaging of the zebrafish oral mucosa (Author response image 1).More nuanced interpretations of the *eda* and *fgf8a* mutant phenotypes.

Reviewer #1 (Evidence, reproducibility and clarity (Required)):The authors describe and characterize the touch system in zebrafish as a new model to study MC development and maintenance. The manuscript is very well written, and the experiments are carefully executed and beautifully illustrated. This study addresses the origin of zebrafish MCs, shows that they are innervated by somatosensory neurons and that they share molecular properties with mammalian MCs. In addition, the authors developed transgenic lines that allow them to study MCs in vivo.Genetic lineage tracing shows that zf MCs are derived from the epidermis as in mouse and not from neural crest cells, as described for avian MCs. In addition, longevity and turnover of murine MCs was controversial. Here, the authors show that zf MCs constantly turnover and that the distribution and turnover rate in the trunk depends on underlying scales. They show that the loss of scales in eda mutants leads to a decrease in MC production and an increase in MC death showing that scales are required for MC production and maintenance. Using a specific fgf8 mutant allele that causes an increase in Fgf signaling and an increase in scale size they demonstrate that scales are sufficient to induce MCs.In summary, this manuscript is a rigorous and beautiful characterization of MCs development and maintenance. The authors demonstrate that zebrafish MCs share many characteristics with mammalian MCs. The generation of MCs specific transgenic lines, coupled with existing transgenic lines that label somatosensory cells and cells in the scales sets the stage for detailed further analyses. For example, using these tools one can now study how the size of the MC progenitor domain is controlled, if progenitors migrate and what the identities of the molecular signals from the nerves and scales to progenitors and differentiated MCs are.Minor comments:Line 71: Why is the heterogeneity a limitation? Couldn't it also exist in zebrafish?

Thank you for raising this question. The limitations are meant to refer to current limitations of the rodent system and demonstrate an opportunity for a new model system to complement the rodent system. We have rephrased this section to better articulate this point.

Introduction:

“While this system has been useful for understanding many aspects of MC development and function, the rodent system also has several significant limitations that warrant additional models to improve the understanding of MCs.”

We also added the following to the discussion: “As the majority of the analyses completed here focus on MCs found in the trunk epidermis, it will be intriguing to determine whether all MCs in different skin compartments in the juvenile and adult zebrafish share similar molecular, cellular, and functional properties.”

Line 295: The authors write: 'Thus, our observations indicate that the decrease in MC cell density in eda mutants is likely due to both reduced MC production and increased MC turnover'.It should say: '.. increased MC loss'. In mutants the MCs show poor turnover. I believe the term 'turnover' implies that the cells are being replaced, which is only partially happening here.

Thank you for the clarification. We agree with the reviewer and have changed the wording from “turnover” to “loss” in lines 295 and 301.

Line 301: 'The authors state: 'these data suggest that Eda signaling is required for MC development, maintenance, and distribution along the trunk.'The authors do not show any data that Eda signaling is involved in MC development but only that scales are needed. The MC inducing signals from scales to the epidermis could be independent from Eda signaling. Please rephrase.Please discuss that not all MC specification/development depends on scales. Even in the scale-less eda mutants some MCs form (as in the inter scale regions in wt?) and even turnover. Do scales secrete a signal that increases proliferation of existing MC progenitors but scales do not affect specification?

We respectfully disagree with the reviewer on the interpretation of these results. Our experimental manipulation (examination of *eda-/-* vs. sibling controls) only allows us to conclude that Eda signaling - either directly or indirectly - is required for these processes along the trunk. To determine whether signaling from scales is required would require identification of the signal(s) and/or loss/ablation of scales independent of Eda. We have rephrased the results to more clearly state our interpretation. The corresponding portion of the discussion now reads “Further investigations are required to determine whether Eda signaling directly regulates the differentiation of MC progenitors. Alternatively, since eda mutants lack scales (Harris et al., 2008) and have decreased epidermal innervation (Rasmussen et al., 2018), MC development may require scale-derived and/or somatosensory neuron-derived signals.”

Line 320: The authors describe that the fgf8 allele leads to a redistribution of MCs. Is it really a redistribution, or is it ectopic induction or expansion of existing progenitors? Redistribution implies that the expansion is due to a loss of MCs in another region, which I do not see in the data.

Thank you for raising this point about the potentially poor wording choice relating to “redistribution”. We do not yet know whether the distribution of MCs in *fgf8a* mutants reflects a redistribution, ectopic induction, or expansion of existing progenitors (these are excellent ideas for future studies). Thus, in response to the reviewer’s comment, we have changed the heading for this results section to “The MC pattern is not predetermined along the trunk” and concluded the section as follows: “... the distribution of MCs tracked with the altered scale size and shape in the mutants, suggesting the MC pattern is not predetermined within the trunk skin compartment (Figure 9E-H).”

Figures:- Figure 1, panels B-C': EM images are very dark and difficult to see. Letter 'a' is on top of the axon, maybe move to the side and pseudo-color different structures.

In response to these suggestions, we have adjusted the brightness and contrast to lighten the TEM images in Figure 1B-C’ as much as possible. We also moved the ‘a’ off to the side in Figure 1B’ to make the axon more visible. In response to Reviewer #3’s comments (see below), we also added an additional TEM image in the new Figure 1—figure supplement 1 that has presumptive keratinocytes and an MC differentially pseudo-colored. We hesitate to pseudo-color the cells/structures in Figure 1B-C’ for fear of obscuring the underlying TEM images.

- Figure 1, panel D: very difficult to see the magenta axons in the cartoon. Please enlarge and make brighter.

We agree that this needed improvement. In the revised Figure 1D, we made the axons clearer and illustrated the different types of MC-axon associations we observe in Figure 2. We also refer the reader back to this figure in the corresponding axon innervation results section.

- Figure 2, panels A and D: keeping the same antibody stainings in the same color would help with visualization. Matching the bar plots in panel C would be even nicer.

Thank you for the suggestion. The revised Figure 2 now has a consistent color scheme.

- Figure 2, panel C: please identify in the legend if the error bars are SD, SEM or other.

These error bars represent 95% confidence intervals. This information has been added to the figure legend.

- Figure 2, panels G and H: MCs are in cyan in the image, but green in the legend.

This has been corrected.

- Figure 3: include percentages and total number in the image instead of the legend.

The numbers and percentages have now been added to the Figure 3 panels. We have left them in the figure legend for clarity on what was scored.

- Figure 6, panel B: which part of the eye is being depicted?

Thank you for the question. We imaged the corneal epithelium above the lens. This has been clarified in the appropriate parts of Figures 6 and 8 and the corresponding figure legends.

- Figure 6, panel F: please provide error bars and statistics to show that the operculum has a higher density of MCs.

Thank you for the suggestion. In response to the comment, we revised Figure 6F by: (1) increasing the sample size; (2) replotting the data as boxplots rather than bar graphs; and (3) including the results of a one-way ANOVA.

- Figure 7, panels F-H: for simple linear regression, please also provide F and p values.

Thank you for the suggestion. This information has been added to the figure legend.

- Figure 8, panel D: colors for SL do not follow a scale, very hard to understand which is which.

In response to the reviewer’s suggestion, we tried numerous different color palettes. However, we were unable to find a color palette that allowed us to distinguish individual points as well as the rainbow palette used in Figure 8D. Thus, after careful consideration, we have elected to keep the original palette here. For consistency, we have used the same palette in the revised Figure 8–figure supplement 1D and Figure 9–figure supplement 1D.

Methods:- Line 472: the word "sex" should be used instead of "gender".

Thank you for the correction. This is fixed in the revision.

- Image analysis, line 593. Please provide a more detailed explanation or describe the ImageJ macro used for the analysis.

Our ImageJ macro has been fully annotated and is provided as Figure 2—source code 1 in the revision. The corresponding methods section has also been updated to clarify the methodology.

Reviewer #1 (Significance (Required)):Soft touch is perceived by Merkel cells (MCs). How MCs develop and are maintained is not well understood because MC development is difficult to study in mammals due to their in utero development. The authors describe and characterize the touch system in zebrafish as a new model to study MC development and maintenance. The study demonstrates that the zebrafish touch system shares many characteristics with its mammalian counterpart, namely its developmental origin, innervation and molecular characteristics. In contrast to mammals, zebrafish transgenic lines that the authors generated, allow the in vivo analysis of Merkel Cell specification, development and maintenance. Therefore this study is the foundation for future detailed cellular and molecular analyses of the touch sensory system and will be of interest to developmental biologists studying stem cells, regeneration and aging, as well as neuroscientists.

We thank the reviewer for their positive assessment of the manuscript.

Reviewer #2 (Evidence, reproducibility and clarity (Required)):This is a very nice and straightforward paper characterizing mechanosensory Merkel cells in the zebrafish skin. The paper uses a number of criteria, based on our knowledge of Merkel cells in mammals, to identify a population atoh1a expressing cells, with neurosecretory granules and actin rich microvilli as Merkel cells in the zebrafish skin. The authors have used existing transgenic lines and developed some of their own, described in this paper, to follow the development of Merkel cell in zebrafish. They show that Merkel cells are derived from basal keratinocytes not neural crest cells. They have region specific densities that influenced by underlying structures like scales and fin rays. They go to show that Ectodysplasin signaling promotes Merkel cell development in the trunk skin but not above the eye or operculum. Reduction of Merkel cells in eda mutants suggest that Eda signaling is required for their development and maintenance. Finally they show that alteration of zebfrafish scale pattern using a mutant with exaggerated fgf8a expression also alters merkel cell distribution.The data presented is clear and the conclusions are supported by their observations.I have no significant issue with the paper as is.Reviewer #2 (Significance (Required)):This study will serve as an excellent basis for future work looking at studies of Merkel cell development and function in fish. Though Merkel cells have been studied in mammals, establishing a zebrafish model for their study will help overcome many barriers that make their analysis difficult in mammals.

We thank the reviewer for their positive assessment of the manuscript.

Reviewer #3 (Evidence, reproducibility and clarity (Required)):In this manuscript, Brown et al. (2022) seek to characterize and address fundamental questions regarding the development and dynamics of Merkel cells (MCs) in zebrafish (*Danio rerio*). The authors utilize a diverse and complementary suite of methods to characterize presumptive MCs in the epidermis of adult zebrafish, including electron microscopy, novel transgenic lines, confocal imaging, and various immuno- and non-immunohistological staining techniques. These studies demonstrate that zebrafish MCs share many features with vertebrate (including mammalian) MCs, particularly regarding morphology/structure, putative functions, genetic markers, and bodily distribution.After establishing the identity of zebrafish MCs, the authors employ lineage tracing and cell tracking analyses to determine that trunk MCs derive from basal keratinocytes and exhibit regular cell turnover. Finally, the authors examine how trunk scales may affect MC development by using established scale mutants. These results show that the presence/absence of scales influences trunk MC development, while scale characteristics (e.g. shape, size) change the distribution of MCs.Major comments:The key conclusions of the manuscript are convincing, however, several points should be addressed by the authors.• Throughout the manuscript, the authors make general claims about zebrafish MCs (zMCs) based on the evidence collected. Yet, most of this evidence (particularly claims about MC turnover, development, structure) comes from examination and experimentation of a specific MC population: trunk MCs located in the scale epidermis. The authors remark upon mammalian MC diversity (lines 73-74) and go on to highlight the diversity of MCs throughout the adult zebrafish (Figure 6), which have differing densities and distribution patterns. Any statements that suggest all zebrafish MCs share certain qualities/features should be carefully considered given the evidence presented.

Thank you for raising this important point. We have added wording in the results and discussion to clearly articulate that the majority of our analyses and conclusions are based on trunk MCs:

Results:

“Anticipating the conclusion of our analysis below, we shall hereafter refer to the epidermal atoh1a+ cells as MCs, with the majority of the analyses completed on trunk MCs unless stated otherwise.”

Discussion:

“As the majority of the analyses completed here focus on MCs found in the trunk epidermis, it will be intriguing to determine whether all MCs in different skin compartments in the juvenile and adult zebrafish share similar molecular, cellular, and functional properties.”

• In the manuscript, the authors validate several markers for the identification of zMCs based upon known mammalian markers (e.g. atoh1a, sox2, piezo2, SV2, 5-HT, and AM1-43; Figures 1-3). Yet, another well-known marker for MCs (CK8) is not addressed (Moll, 1995; Moll, 2005). One zebrafish ortholog for CK8 is krt4, a transgene successfully employed in this study to label keratinocytes. Do zMCs express krt4 or other mammalian MC keratins? Answering this question or addressing this discrepancy would further strengthen the authors claims that these cells are bona fide zMCs.

We agree with the reviewer that (1) identification of a keratin(s) that distinguishes MCs from other epidermal cell types in zebrafish would be an excellent reagent; and (2) readers familiar with the mammalian MC literature may similarly wonder why this was not addressed in the manuscript. Indeed, we had considered whether we could identify homologs of CK8, CK20 or other mammalian MC keratins that would label zebrafish MCs. However, despite the confusing nomenclature that would indicate otherwise, the zebrafish keratins share more homology with each other than the corresponding mammalian proteins (Ho et al., 2022; PMID: 34991727). Our revised results section now includes the following to clarify this point: “For example, keratins, most notably keratin 8 and keratin 20, have been used extensively as markers of mammalian MCs (Moll et al., 1995, 1984). However, zebrafish keratins have undergone extensive gene loss and duplication and are not orthologous to mammalian keratin genes (Ho et al., 2022). Thus, we considered alternative molecular markers to label zebrafish MCs.”

• The authors utilize a previously validated eda mutant line to see if ectodysplasin signaling affects zMC development. While the results of these experiments are convincing, the authors need to make clear whether they are claiming that scales, scale-derived Eda signaling, or Eda signaling alone dictate trunk MC development. It appears that there is some conflation of these ideas, particularly with line 306 ("blocking dermal appendage formation inhibited MC development" is a different claim from 'blocking Eda signaling inhibited MC development'). One way to make this differentiation would be to perform a similar experiment as detailed in Xiao et al. 2016: using a Shh agonist in eda mutants. If scale-specific signals are required in addition to Eda, we would expect to see similar MC densities and patterns in both Shh agonist-treated and non-treated eda mutants.

We agree that our interpretation of these results could have been more clearly articulated in our initial submission. As discussed above in response to Reviewer #1, we do not yet know whether Eda signaling directly or indirectly influences MC development. We have revised the results section to clarify our interpretation of the results as follows: “Together, these data suggest that either Eda signaling, or a scale-derived signal, is required for MC development, maintenance, and distribution along the trunk. Further studies are required to determine the specific scale-derived signal that regulates MC development in the trunk.”

The suggestion of using a Shh pathway agonist in *eda* mutants to attempt to rescue MC differentiation similar to Xiao et al. 2016 is an interesting one. To our knowledge, experiments validating the Smo agonist used by Xiao and colleagues (Hh-Ag1.5) in zebrafish have not been published. We also note that activation of Shh signaling by heat-shock induction of *shha* expression during squamation led to kyphosis and epidermal migration off of the trunk (Aman et al., 2018; PMID: 30014845). Thus, we respectfully suggest that distinguishing between the various possibilities downstream of Eda is beyond the scope of the current manuscript. We have added a discussion point along these lines: “Further investigations are required to determine whether Eda signaling directly regulates the differentiation of MC progenitors. Alternatively, since eda mutants lack scales (Harris et al., 2008) and have decreased epidermal innervation (Rasmussen et al., 2018), MC development may require scale- and/or somatosensory neuron-derived signals. Finally, we note that trunk MCs are not completely absent in eda mutants, suggesting that a subset of MCs develop independent of Eda signaling.”

• Throughout the manuscript, the authors use subjective language (e.g. line 106). While this reviewer does not wish to suppress or alter the authors' voices, careful consideration should be used when employing these types of descriptors. Furthermore, the authors use suggestively quantitative language inappropriately or unjustifiably. For example, in line 221, the authors use "extensive" when describing the co-labeling between atoh1a+ MCs and lineage-traced basal keratinocytes; the percentage of co-labeled cells ranged from 29-32%. Other quantitative descriptors such as "frequently" (line 171) or "uniform" (line 249) describe various features or phenomena without quantification in figures or supplements.

Thank you for this comment. We have paid careful attention to our subjective/statistical language in the revision. Regarding the usage of “uniform” - we have added the wording “relatively uniformly” to descriptions and a statement that our term “uniform” was not specifically quantified. Although the uniform appearance was not specifically quantified, we believe this provides an accurate description of the MC localization pattern in certain skin compartments.

Example word change in Results:

“For example, MCs were distributed relatively uniformly across the eye, although this spatial pattern was not specifically quantified.”

• In the lineage tracing experiments (Figure 4), the authors note that "recombination is not complete" (lines 1016-1017) to explain why not all zMCs express the basal keratinocyte lineage marker. While this idea could be supported by Figure 4-figure supplement 1, one could postulate that zMCs are derived from multiple progenitor lineages. Using the basal keratinocyte lineage tracing validation, the authors could in theory calculate a "recombination efficiency" of this transgenic line and determine approximately the percent of zMCs they 'lose' as a result. Otherwise, the authors could perform other experiments to support the claim that zMCs derive from basal keratinocytes. For example, could the authors photoconvert basal keratinocytes at 1 dpf and see how many derived MCs are still photoconverted later? Could they do this photoconversion experiment with neural crest cells? Could they ablate neural crest cells and determine if MC number is affected? These additional experiments are not necessarily required for publication, but some explanation of the unexpectedly low percentage of basal keratinocyte lineage marker-labeled MCs would suffice.

We thank the reviewer for raising this important point and the suggestion of calculating a “recombination efficiency”. We note that Cre responsive transgenes are far from a perfect technology in zebrafish as recently characterized by Lalonde et al. (2022; PMID:35582941). In response to the reviewer’s comment, we added an estimate of the recombination efficiency to Figure 4 (panels E, G, H). Importantly, a comparison between the recombination efficiency and percentage of MCs labeled by the basal keratinocyte Cre tracing was not significantly different. Our revised results section reads as follows: “After raising 4-OHT-treated animals to adulthood, we observed variable (2-81%) co-labeling between the basal keratinocyte lineage trace and MC reporters (Figure 4D’,F). We note that our lineage tracing strategy did not label all basal keratinocytes (Figure 4D; Figure 4—figure supplement 1), suggestive of incomplete Cre-ERT2 induction and/or transgene recombination. Consistent with the latter possibility, a recent analysis demonstrated *Tg(actb2:LOXP-BFP-LOXP-DsRed)* has a low recombination efficiency compared to other Cre reporter transgenes (Lalonde et al., 2022). To estimate the local recombination efficiency in imaged regions, we thresholded the DsRed channel and calculated the fraction of skin cells labeled (Figure 4E). Importantly, the proportion of MCs labeled by the basal keratinocyte lineage trace was not significantly different from the local recombination efficiency (Figure 4G-H). These observations support a basal keratinocyte origin of most or all zebrafish MCs.”

• The authors use appropriate statistics and have sufficient replicates when this information is presented. Yet, the presence or absence of these data is not consistent within figure captions. The authors must ensure that they provide the N of adults and scales (when appropriate), the SL range of adults, and transgenic lines used. Statistics are missing in some figures (for example: Figures 4E, 5D, 5E, 6F, 8S-1E, 9E-H) where it would be appropriate to include them. In some figures, the N changes over time (example: 5D, 5E); an explanation in the 'Methods' section would suffice.

Thank you for noting the need for additional statistics. We have added statistics to the above figures. For Figure 9E-H, we have not added additional statistics. Figure 9E-H serve to graphically visualize differences. We show statistical differences in Figure 9—figure supplement 1 for scale area, aspect ratio, and Feret’s diameter. We have added an explanation related to Figure 5D,E in the methods section: “Animals that died over the course of the experiment were excluded from further analysis.”

Minor commentsWhile the authors present an extensive argument for their claims, addressing these additional comments would further strengthen their story.• Are zMC nuclei lobulated? This ultrastructure characteristic seems to be common in MCs (Chew & Leung, 1994; Tachibana & Nawa, 2002; Moll, 2005; Boulais, 2009).

We have not observed any lobulation of the MC nuclei by TEM, nor was this commented on in the TEM studies of Whitear and colleagues in other teleosts (Lane and Whitear, 1977; PMID: 198137; Whitear, 1989; PMID: 2510796). Nevertheless, we cannot rule out the possibility that serial sectioning or other high resolution analysis of the nuclear shape may reveal such features. In response to the reviewer’s comment, we have added the following paragraph to the discussion: “While our characterization revealed substantial similarities between mammalian and zebrafish MCs, we did observe anatomical differences in line with previous ultrastructural characterizations of teleost MCs (Lane and Whitear, 1977; Whitear, 1989). For example, the nuclei of mammalian MC are commonly lobulated (Boulais et al., 2009; Cheng Chew and Leung, 1994; Moll et al., 2005; Tachibana and Nawa, 2002). While we did not observe lobulation of zebrafish MC nuclei by TEM, we cannot rule out that serial sectioning or high-resolution reconstruction of nuclear shape would reveal lobulation. Mammalian MCs typically localize adjacent to basal keratinocytes (Boot et al., 1992; Cheng Chew and Leung, 1994; Fradette et al., 1995; Mihara et al., 1979; Moll et al., 1996; Smith, Jr, 1977), whereas zebrafish MCs appear in upper strata, typically beneath the periderm (Figure 1D,G’’). As the majority of the analyses completed here focus on MCs found in the trunk epidermis, it will be intriguing to determine whether all MCs in different skin compartments in the juvenile and adult zebrafish share similar properties.”

• In Figure 3C and 3", the authors show that AM1-43 labels zMCs. Yet, this technique should also stain sensory axons that associate with MCs (Meyers, 2003). Are axons also stained? Other positive controls for the stains could be useful as a supplement.

The reviewer is correct that Meyers et al., (2003; PMID: 12764092) report AM1-43 staining of neurites that innervate MCs in the whisker follicle. However, they did not report similar staining of neurites innervating touch dome MCs. In murine hairy skin, the related styryl dye FM1-43 appears to most prominently stain MCs and hair follicle-associated lanceolate endings (Banks et al., 2013 PMID: 23440964; Villarino et al., 2022 preprint DOI: 10.1101/2022.05.26.493600). Our revised legend for Figure 3 now includes the following: “AM1-43 has been reported to stain neurites innervating MCs in murine whisker vibrissae (Meyers et al., 2003). However, our AM1-43 staining regiment did not strongly label cutaneous axons, although we cannot exclude low levels of staining.”

All of the stains used in our original Figure 3 have been previously validated in zebrafish, which we have more clearly stated and cited in the corresponding results section of the revision. Because these reagents have all been previously validated and our staining patterns are consistent with the literature, we respectfully suggest that positive controls would add little value to the current manuscript. Nevertheless, in response to the reviewer’s comment, we confirmed our *piezo2* FISH staining using an independent method (a *piezo2* HCR probe). We have included these HCR results as the updated Figure 3D and moved the original Figure 3D to Figure 3—figure supplement 1.

• In Figure 7, the authors argue that as scales develop, MC density increases with scale area. Did the authors compare MC densities of differently-sized scales at the same age? Is fish SL/age a potential confound in the interpretation of these data?

Thank you for the suggestion. In response to the reviewer’s comment, we have replotted the data in Figure 7G,H for animals in the range 8-10 mm SL in Figure 7—figure supplement 1. We have revised the corresponding results section as follows: “The density and number of MCs positively correlated with scale area (Figure 7G,H), although this trend was less pronounced at stages less than 10 mm (Figure 7—figure supplement 1)”. As discussed above in response to Reviewer #1’s suggestion, we also now report *F*-statistics and *P*-values for the linear regressions in the figure legends.

• The authors claim that squamation begins at ~9 mm SL (line 268), prior to which MCs were "rare" in the epidermis (supported by data in Figure 7F). However, Figures 8A and 8G suggest that MCs are not rare prior to squamation/9 mm SL. Are these data in conflict?

Thank you for raising this observation. We do not believe these data are in conflict. Figure 8A and B show images of fish 8.8-8.9 mm SL, immediately prior to squamation. MCs appear about the same time as scales develop but the exact timing varies between animals. To further strengthen this section of the manuscript, we now include quantification of the density of trunk MCs at various stages prior to 9 mm SL (new data added to the developmental timeline in Figure 7F). These data are consistent with our initial interpretation. In the revised results section we clarify this as follows: “Using reporters that label MCs and scale-forming osteoblasts, we rarely observed MCs in the epidermis prior to 8 mm SL (Figure 7B, F). Between 8-10 mm SL, MCs appeared at a low density along the trunk (Figure 7F). MC density rapidly increased from 10-15 mm SL, a period of active scale growth (Figure 7C-F).”

• In Figure 6B-E, the panels are incorrectly labeled as "atoh1a:nls-Eos" (figure caption and fluorescence localization show they are atoh1a:Lifeact-EGFP).

The low magnification panels were correctly labeled as *atoh1a:nls-Eos*. The insets showed *atoh1a:Lifeact-EGFP* as described in the figure legend. We apologize for the confusion and poor data presentation. We have revised Figure 6 to eliminate the problematic labeling/display.

• Figure 9 panels E-H are not referenced in the main body of the text.

Thank you for pointing this out. Fixed in the revised manuscript.

• In Figure 6, the authors examine MC densities in the tail, but do not quantify changes here with eda mutants as they did for other regions (eye, operculum) in Figure 8. Why was this region not examined?

We have clarified this point in the revised results section as follows: “eda mutants lack fins at the stages analyzed (Harris et al., 2008) precluding analysis of these regions in the homozygous mutants.”

• The authors do a good job in detailing the current literature regarding MCs, however, two missing areas are noticeable: (1) there is no mention of mammalian MCs that reside in the oral mucosa (Hashimoto, 1972) or whether they exist in zebrafish, and (2) no mention of Merkel-like cells (Halata, 2003) and why the cells in this paper are or are likely not Merkel-like cells.

Thank you for the suggestions. Regarding the first point, we revised the introduction to reference (Hashimoto, 1972) as follows: “...vertebrates have diverse types of skin and MCs are found in both hairy and glabrous (non-hairy) skin, as well as mucocutaneous regions such as the gingiva and palate (*Hashimoto, 1972*; Lacour et al., 1991; Moayedi et al., 2021).” We also imaged the mucosal tissue along the roof palate of the adult mouth and identified *atoh1a+* cells (see Author response image 1). Close examination of the *atoh1a:Lifeact-EGFP* signal revealed these cells have a spherical morphology and extend short processes similar to the MCs described across the body regions examined in Figure 6. However, as the microvillar morphology of the palatal *atoh1a+* cells is not identical to those identified in other skin regions, we hesitate to call these MCs without performing additional in-depth analyses. We feel that inclusion of these data in the manuscript could distract the reader from the main focus of our study, therefore we have included them here:

**Author response image 1. sa2fig1:** *atoh1a*+ cells in the adult oral epithelium. (**A,B**) Low- (**A**) and high-magnification (**B**) confocal micrographs of oral roof palate epithelium in an adult expressing reporters for keratinocytes (*Tg(krt4:DsRed)*) and *atoh1a*-expressing cells (*Tg(atoh1a:Lifeact-EGFP)*). (**B’**) Reconstructed cross section along the yellow line in B showing two *atoh1a+* cells in the upper strata of the oral epithelium. Scale bars: 50 µm (**A**) and 10 µm (**B,B’**).

Regarding the second point, we have added the following sentence to the first paragraph of the discussion: “Second, zebrafish MCs extend numerous short, actin-rich microvilli and complex with somatosensory axons, classic morphological hallmarks of MCs (Mihara et al., 1979; Smith, Jr, 1977; Toyoshima et al., 1998). Our morphological observations support the interpretation that these cells are MCs rather than Merkel-like cells, which lack axon association and microvillar processes (reviewed by Halata et al., 2003).”

• It may help readers understand MC morphology in context if the authors include a larger picture of the TEM data that highlights the drastic difference in ultrastructure between MCs and neighboring keratinocytes.

Thank you for the suggestion. We added a new figure (Figure 1—figure supplement 1) to the revised manuscript that contains an additional TEM image that we believe illustrates the different morphologies of keratinocytes and MCs. We hope this will help the reader contextualize the morphology and position of MCs within the zebrafish epidermis. This is now referenced in the first results section as follows: “The cells appeared relatively small and spherical with a low cytoplasmic-to-nuclear ratio compared to neighboring keratinocytes (Figure 1B,C; Figure 1—figure supplement 1) …”

Reviewer #3 (Significance (Required)):The current manuscript provides significant advancements in various biological fields and research communities. For researchers that utilize zebrafish as a model organism, these findings present a new cell type along with novel and essential genetic tools for study. These developments open the possibilities to further understand MCs, their roles in somatosensory function, mechanisms of cell type diversification, and to engage in translational research. For those already researching MCs, this manuscript shows that fundamental questions regarding MC functioning can be rigorously addressed with a new model that can fill the methodological limitations imposed by mammalian biology. Indeed, the authors do a thorough job of introducing and contextualizing our knowledge of MCs and any outstanding gaps. The authors then sit their findings comfortably alongside previous works, largely supporting those findings, and take the extra step to address MC controversies/matters of debate. This technique of supporting the current literature and then uplifting it with new findings makes this work even more impressive. Various audiences will find value in this manuscript, including but not limited to those that study epidermal cell types, the development and influence of skin appendages, somatosensation and sensory disorders, developmental biology, and Merkel cell carcinoma.

We thank the reviewer for their positive assessment of the manuscript.